# DECOUPLED MEANFLOW: TURNING FLOW MODELS INTO FLOW MAPS FOR ACCELERATED SAMPLING

**Kyungmin Lee**    **Sihyun Yu**    **Jinwoo Shin**
KAIST

## ABSTRACT

Denoising generative models, such as diffusion and flow-based models, produce high-quality samples but require many denoising steps due to discretization error. Flow maps, which estimate the average velocity between timesteps, mitigate this error and enable faster sampling. However, their training typically demands architectural changes that limit compatibility with pretrained flow models. We introduce *Decoupled MeanFlow*, a simple decoding strategy that converts flow models into flow map models without architectural modifications. Our method conditions the final blocks of diffusion transformers on the subsequent timestep, allowing pretrained flow models to be directly repurposed as flow maps. Combined with enhanced training techniques, this design enables high-quality generation in as few as 1–4 steps. Notably, we find that training flow models and subsequently converting them is more efficient and effective than training flow maps from scratch. On ImageNet 256×256 and 512×512, our models attain 1-step FID of 2.16 and 2.12, respectively, surpassing prior art by a large margin. Furthermore, we achieve FID of 1.51 and 1.68 when increasing the steps to 4, which nearly matches the performance of flow models while delivering over 100× faster inference.[1]

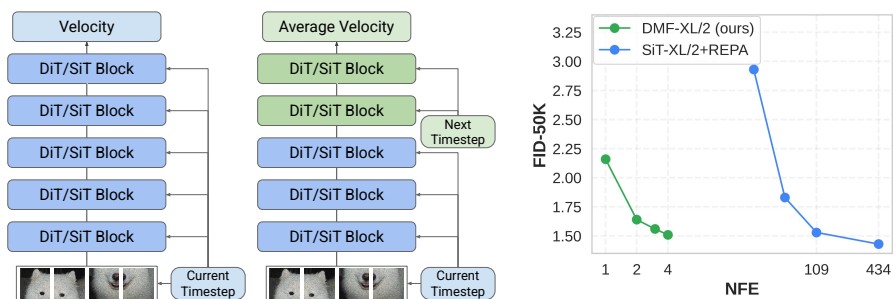

Figure 1: **Accelerating diffusion transformer via Decoupled MeanFlow.** (Left) Our model, Decoupled MeanFlow (DMF), converts a flow model into a flow map by decoding the intermediate representation with next timestep $r$, while preserving the original architecture. (Right) Fine-tuning DMF-XL/2 to predict average velocity (Geng et al., 2025a) significantly accelerates the sampling speed of flow model (SiT-XL+REPA; Yu et al. 2025), while maintaining the performance.

## 1    INTRODUCTION

Diffusion models (Sohl-Dickstein et al., 2015; Ho et al., 2020; Song et al., 2021) and flow models (Lipman et al., 2023; Albergo et al., 2023) have emerged as effective and scalable approaches for generating high-quality visual data, including images (Ramesh et al., 2021; Saharia et al., 2022; Rombach et al., 2022; Esser et al., 2024) and videos (Blattmann et al., 2023; Brooks et al., 2024; Polyak et al., 2024; Wan et al., 2025). Despite their success, improving sampling efficiency remains a key challenge, since producing high-quality samples typically requires many denoising iterations.

---

[1]Model weights and code available at `https://github.com/kyungmnlee/dmf`.

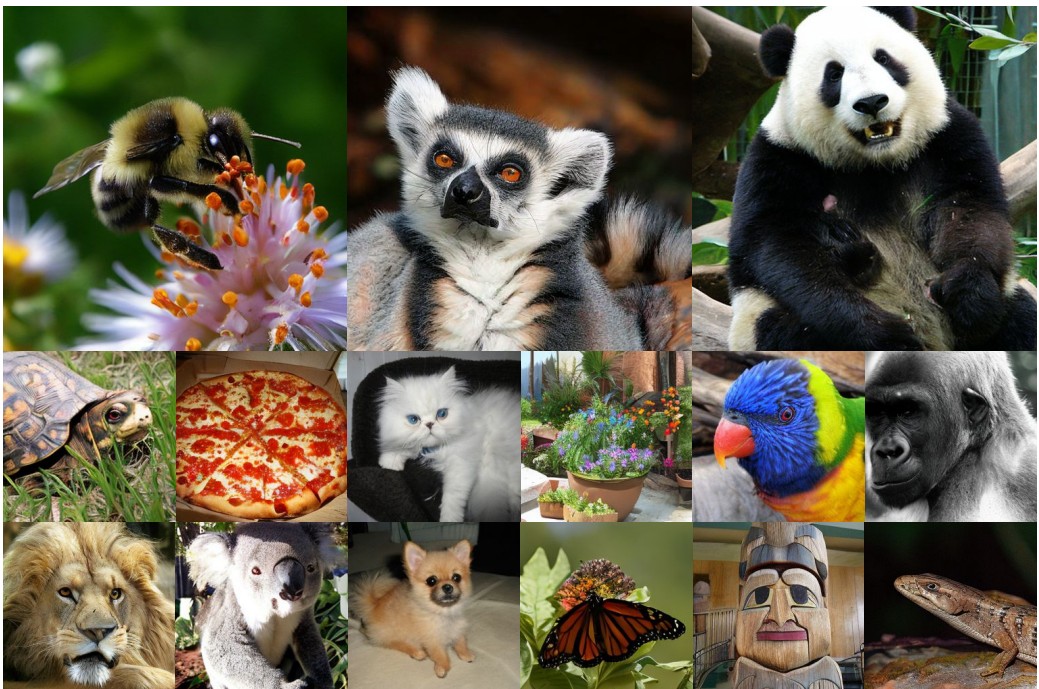

Figure 2: **Qualitative examples.** Selected samples from our DMF-XL/2+ models trained on ImageNet 512×512 (top row) and ImageNet 256×256 (bottom row) using NFE = 1 (left), 2 (middle), 4 (right).

To address this inefficiency, recent research has explored principled methods for designing diffusion models with fewer sampling steps. Consistency models (Song et al., 2023; Song & Dhariwal, 2024; Lu & Song, 2025; Geng et al., 2025b; Peng et al., 2025; Heek et al., 2024) enforce consistency between the denoised outputs from adjacent timesteps, enabling 1- or 2-step generation. While promising, these models struggle to scale effectively beyond two steps (Sabour et al., 2025). Another line of work focuses on learning *flow maps*, which models the average velocity between two timesteps (Kim et al., 2024; Geng et al., 2025a; Sabour et al., 2025; Boffi et al., 2024; 2025). In particular, MeanFlow (Geng et al., 2025a) provides a principled generalization of flow matching, showing that flow maps can achieve performance comparable to standard flow models.

While MeanFlow demonstrates the potential of flow maps, its architectural design remains under-explored. Specifically, it integrates the next timestep information throughout the diffusion transformer (Peebles & Xie, 2023), implicitly assuming that both encoder and decoder must rely on it. Yet, this assumption may be unnecessary: the encoder's role is to extract a representation from noisy inputs, where incorporating future timestep information may add little value. In contrast, the decoder is precisely where the next timestep should matter, as it governs how the model predicts future states.

In this paper, we introduce *Decoupled MeanFlow (DMF)*, a simple approach that transforms pretrained flow models into flow maps without altering their architecture. Our key idea is to decouple the diffusion transformer into encoder and decoder components: the encoder processes the current timestep, while the decoder incorporates the next timestep (see Fig. 1). This formulation avoids unnecessary architectural modifications while retaining compatibility with existing flow models.

Our design is inspired by recent works to rethink the representational structure of generative models, such as representation alignment (Yu et al., 2025), regularization (Wang & He, 2025), and masked modeling (Li et al., 2024). We hypothesize that the next timestep information is irrelevant during representation encoding and only necessary in decoding for learning flow maps.

Our approach offers several advantages. DMF can fully reuse the pretrained flow model without architectural modification. As a result, any flow model can be seamlessly repurposed as a flow map model. We show that even without fine-tuning, the converted models can produce high-quality samples, often surpassing their original flow model counterparts (Fig. 3b). Moreover, fine-tuning only the decoder substantially accelerates sampling while preserving quality (Fig. 3c).

Beyond the method itself, we provide a comprehensive analysis of Decoupled MeanFlow. We show that fine-tuning from pretrained flow models not only yields higher performance than training flow maps from scratch, but also requires fewer training compute, making our approach more efficient (Fig. 4). Moreover, our study reveals that the representational capacity plays a critical role in learning effective flow maps, highlighting the importance of encoder–decoder decoupling (Tab. 1 and Tab. 2).

Quantitatively, Decoupled MeanFlow sets a new state-of-the-art in few-step generative modeling. On ImageNet 256×256, our model achieves a 1-NFE FID=2.16, surpassing prior few-step diffusion models (Geng et al., 2025a; Frans et al., 2025; Zhou et al., 2025) and other approaches such as GANs (Sauer et al., 2022) and normalizing flows (Gu et al., 2025). When increasing the number of steps to 4, DMF reaches an FID=1.51, matching the performance of flow models (Yu et al., 2025) that require over 100× more computation during inference (see Tab. 3). Our approach successfully applies to higher resolution, for example, on ImageNet 512×512, we achieve 1-NFE FID=2.12, and 2-NFE FID=1.75, outperforming the prior art sCD (Lu & Song, 2025) (see Tab. 4).

## 2 PRELIMINARIES

**Flow models.** Flow models (Lipman et al., 2023; Albergo et al., 2023) (or diffusion models (Ho et al., 2020; Song et al., 2021)) consist of a forward process that adds noise to the data, and a reverse process that gradually denoise noisy input to clean data. Formally, given a data $\mathbf{x}_0 \sim p_{\text{data}}$ and noise $\epsilon \sim \mathcal{N}(\mathbf{0}, \mathbf{I})$, the forward process at time $t \in [0, 1]$ is given by $\mathbf{x}_t = \alpha_t \mathbf{x}_0 + \sigma_t \epsilon$, where $\alpha_t$ and $\sigma_t$ are coefficients that satisfy $\alpha_0 = \sigma_1 = 1$ and $\alpha_1 = \sigma_0 = 0$. Training a flow model $\mathbf{v}_\theta$ is done by predicting the velocity $\mathbf{v}(\mathbf{x}, t) = \mathbb{E}[\alpha_t' \mathbf{x}_0 + \sigma_t' \epsilon | \mathbf{x} = \mathbf{x}_t]$ with flow matching objective as follows:

$$\mathcal{L}_{\text{FM}}(\theta) = \mathbb{E}_{\mathbf{x}_t, t} \left[ \| \mathbf{v}_\theta(\mathbf{x}_t, t) - \mathbf{v}(\mathbf{x}, t) \|^2 \right]. \tag{1}$$

Given the velocity $\mathbf{v}_\theta(\mathbf{x}, t)$, the generative reverse process obtains a sample by solving the probability flow ODE for $\mathbf{x}_t$, *i.e.*, $\mathrm{d}\mathbf{x}_t = \mathbf{v}_\theta(\mathbf{x}_t, t)\mathrm{d}t$. Note that the exact solution of ODE for $\mathbf{x}_t$ from time $t$ to $r$ is given as $\mathbf{x}_r = \mathbf{x}_t + \int_t^r \mathbf{v}_\theta(\mathbf{x}_\tau, \tau)\mathrm{d}\tau$. In practice, since the integration is intractable, we resort to numerical methods, *e.g.*, Euler's method, where the solution $\mathbf{x}_r$ from $\mathbf{x}_t$ is given by $\mathbf{x}_r = \mathbf{x}_t + (r - t)\mathbf{v}_\theta(\mathbf{x}_t, t)$. However, such numerical approaches require a large number of denoising steps or high-order methods (Karras et al., 2022; Lu et al., 2022; 2025) to achieve high-quality samples in order to reduce the time-discretization error.

**Flow maps.** To accelerate the sampling, recent works (Geng et al., 2025a; Sabour et al., 2025; Boffi et al., 2024) propose to learn a *flow map* between two timesteps that accelerate the inference speed via reducing the discretization error. Let $\mathbf{u}_\theta(\mathbf{x}_t, t, r)$ be a flow map of $\mathbf{x}_t$ from $t$ to $r$, and the ODE solver with flow map model be $\mathbf{x}_r = \mathbf{x}_t + (r - t)\mathbf{u}_\theta(\mathbf{x}_t, t, r)$. Then the time-discretization error of the flow map ODE solver can be written as following:

$$\text{Err}(\mathbf{x}_t, t, r) = \left\| \int_t^r \mathbf{v}(\mathbf{x}_\tau, \tau)\mathrm{d}\tau - (r - t)\mathbf{u}_\theta(\mathbf{x}_t, t, r) \right\|^2. \tag{2}$$

One can derive the training objective for the flow map model by minimizing the time-discretization error Eq. (2). Specifically, since the integration is non-achievable, MeanFlow (Geng et al., 2025a) introduced a training objective that minimizes the gradient norm of discretization error:

$$\mathcal{L}_{\text{MF}}(\theta) = \mathbb{E}_{\mathbf{x}_t, r} \left[ \left\| \mathbf{u}_\theta(\mathbf{x}_t, t, r) - \mathbf{v}(\mathbf{x}_t, t) - (r - t)\frac{\mathrm{d}}{\mathrm{d}t}\mathbf{u}_\theta(\mathbf{x}_t, t, r) \right\|^2 \right], \tag{3}$$

where the last term is computed by Jacobian-vector product (JVP) between primal vectors $(\partial_\mathbf{x}\mathbf{u}_\theta, \partial_t\mathbf{u}_\theta, \partial_r\mathbf{u}_\theta)$ and tangent vectors $(\mathbf{v}, \mathbf{1}, \mathbf{0})$ using following identity:

$$\frac{\mathrm{d}}{\mathrm{d}t}\mathbf{u}_\theta(\mathbf{x}_t, t, r) = \frac{\mathrm{d}\mathbf{x}_t}{\mathrm{d}t}\frac{\partial\mathbf{u}_\theta}{\partial\mathbf{x}_t} + \frac{\partial\mathbf{u}_\theta}{\partial t} = \mathbf{v}(\mathbf{x}_t, t)\frac{\partial\mathbf{u}_\theta}{\partial\mathbf{x}_t} + \frac{\partial\mathbf{u}_\theta}{\partial t}.$$

In practice, to eliminate the double backpropagation for JVP and ease the optimization (Lu & Song, 2025; Frans et al., 2025; Geng et al., 2025b), we set $\mathbf{u}_{\text{tgt}} = \mathbf{v} + (r - t)\frac{\mathrm{d}\mathbf{u}_\theta}{\mathrm{d}t}$ and optimize with $\mathcal{L}_{\text{MF}}(\theta) = \mathbb{E}_{\mathbf{x}_t, r}[\|\mathbf{u}_\theta - \text{sg}(\mathbf{u}_{\text{tgt}})\|^2]$, where $\text{sg}$ is a stop-gradient operator. Remark that minimizing Eq. (3) alone cannot make the discretization error to be zero, as it only optimizes the gradient norm to be zero (*i.e.*, first-order condition). Thus, it requires to satisfy boundary condition (*i.e.*, $r = t$), which becomes equivalent to flow matching objective as in Eq. (1).

**Designing flow map architecture.** To encode the timestep into the diffusion model, it is common practice to use positional embedding layer (Ho et al., 2020; Vaswani et al., 2017) that conditions throughout the layers. For instance, Diffusion Transformer (DiT; Peebles & Xie 2023) uses timestep embeddings to modulate the outputs of MLP and attention layers inside the transformer blocks. To implement flow maps, recent approaches (Zhou et al., 2025; Geng et al., 2025a; Sabour et al., 2025) use an extra timestep embedding layer for $r$ and add the embeddings from $t$ and $r$ (or $t - r$) for the condition embedding (see Appendix B.1 for visualization).

**Model Guidance.** Classifier-free guidance (CFG; Ho & Salimans 2022) is a common practice to enhance conditional generation, where it interpolates between conditional and unconditional velocities to control. However, CFG comes at the cost of doubling the inference compute. To reduce the inference cost, *model guidance* (MG; Tang et al. 2025) adjusts a target velocity as following:

$$\mathbf{v}^{\text{tgt}}(\mathbf{x}_t, t, \mathbf{y}) = \mathbf{v}(\mathbf{x}_t, t) + \omega\big(\mathbf{v}_\theta(\mathbf{x}_t, t, \mathbf{y}) - \mathbf{v}_\theta(\mathbf{x}_t, t)\big), \tag{4}$$

where $\mathbf{y}$ is a condition and $\omega \in (0, 1)$ is a model guidance scale. Then, training with MG is done by applying stop-gradient operator to $\mathbf{v}_{\text{tgt}}$ and replace $\mathbf{v}$ with $\mathbf{v}^{\text{tgt}}$ in Eq. (1). Notably, MG is effective when training flow map, which enables high-quality 1-NFE generation (Geng et al., 2025a). We further provide details of model guidance in Appendix E.

## 3 PROPOSED METHOD

### 3.1 DECOUPLED MEANFLOW

Previous works have shown that diffusion and flow models implicitly perform representation learning (Li et al., 2023; Xiang et al., 2023; Chen et al., 2024), and improving the representations further enhances the generation capability (Yu et al., 2025; Wang et al., 2025; Wang & He, 2025). As such, one can reinterpret a flow model $\mathbf{v}_\theta(\mathbf{x}_t, t)$ as the composition of an encoder $f_\theta : \mathcal{X} \times [0, 1] \to \mathcal{H}$ and a decoder $g_\theta : \mathcal{H} \times [0, 1] \to \mathcal{X}$ such that $\mathbf{v}_\theta = g_\theta \circ f_\theta$, where $\mathbf{h}_t = f_\theta(\mathbf{x}_t, t)$ and $\mathbf{v}_\theta(\mathbf{x}_t, t) = g_\theta(\mathbf{h}_t, t)$.

From this perspective, the representation encoded by $f_\theta$ should also matter in learning a high-quality flow map. However, the architectural design of flow maps remains unclear. Previous works (Kim et al., 2024; Zhou et al., 2025; Geng et al., 2025a) follow a straightforward approach that the next timestep $r$ is provided to both the encoder and the decoder. Yet, this design may introduce redundancy: the encoder's task is to extract semantic features from $\mathbf{x}_t$, for which the next timestep $r$ may be irrelevant. Conversely, once the encoder produces $\mathbf{h}_t$, the decoder's role is to predict the average velocity toward timestep $r$, which may no longer require the original $t$.

These observations motivate us to decouple timestep conditioning in the encoder and decoder. Specifically, we propose to drop $r$ from the encoder and $t$ from the decoder, leading to the formulation $\mathbf{u}_\theta(\mathbf{x}_t, t, r) = g_\theta(f_\theta(\mathbf{x}_t, t), r)$. We refer to this architectural design as *Decoupled MeanFlow (DMF)*, as the encoder and decoder are conditioned on different timesteps in a complementary manner. Following Yu et al. (2025), we partition the diffusion transformer into the first $d$ layers as encoder $f_\theta$ and the remaining $\ell - d$ layers as decoder $g_\theta$, where $\ell$ is the total number of blocks. To avoid introducing additional parameters, we reuse the same timestep embedding layer for both $t$ and $r$, preserving the velocity prediction module of the original flow model (see Fig. 1).

**Your flow model is secretly a flow map.** This formulation suggests that any pretrained flow model can be converted into a flow map via DMF. To test this hypothesis, we take the SiT-XL/2+REPA (Yu et al., 2025) pretrained on ImageNet (Deng et al., 2009), convert it into a flow map (DMF-XL/2) without any fine-tuning, and compare their generative performance. We generate 50K samples using the Euler solver without classifier-free guidance (CFG) and evaluate FID (Heusel et al., 2017).

In Fig. 3a, we vary the encoder depth $d$ for DMF with 16 denoising steps. Interestingly, DMF consistently improves as the decoder becomes smaller, and even outperforms the original SiT model in several settings. In Fig. 3b, fixing $d=22$, DMF maintains a clear advantage across denoising steps from 16 to 128. These results confirm that flow models can be transformed into flow maps without any fine-tuning, and that carefully choosing the encoder–decoder split can even yield better generative quality. We further investigate this finding across different flow models (see Appendix F).

**Representation matters for flow map.** We next study how encoder representations influence flow map quality. To this end, we freeze the encoder ($d=22$) and fine-tune only the decoder (and timestep

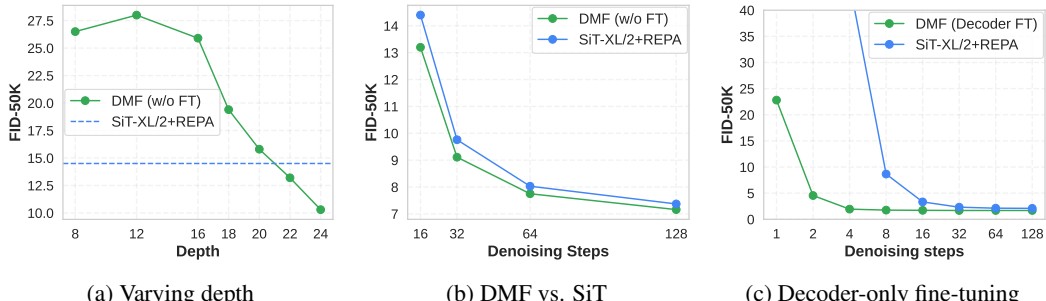

(a) Varying depth        (b) DMF vs. SiT        (c) Decoder-only fine-tuning

Figure 3: **Pretrained flow model as a flow map.** Comparison between pretrained flow model (SiT-XL/2+REPA; Yu et al. 2025) and converted flow map (*i.e.*, see Fig. 1) with FID-50K is reported. (a) Converted DMF without fine-tuning (DMF w/o FT) outperforms SiT-XL+REPA when chosen proper decoder depth. (b) Fixing depth to 22 and varying the denoising steps, DMF w/o FT consistently outperform pretrained SiT-XL/2+REPA. (c) By freezing the encoder and fine-tuning the decoder with flow map loss with guidance, decoder-tuned DMF (DMF Decoder FT) achieves substantial gain in sampling efficiency compared to SiT-XL/2+REPA with CFG.

embeddings layers) using the flow map training objective Eq. (3) with model guidance. We fine-tune SiT-XL/2+REPA for 40 epochs on ImageNet 256×256 resolution.

As shown in Fig. 3c, the decoder fine-tuned DMF achieves substantial efficiency gains: with only 8 denoising steps, it reaches FID=1.76, outperforming the baseline SiT-XL/2+REPA with CFG scale 1.5. This demonstrates that high-quality encoder representations learned by flow models can be effectively transferred to flow maps through DMF. However, we also observe that the 1-step performance remains limited when the encoder is frozen, indicating that the encoder must be jointly optimized to unlock the full potential of 1-step generative modeling.

## 3.2 IMPLEMENTATION

**Training algorithm.** Since our method combines the flow matching (FM) loss Eq. (1) with the MeanFlow (MF) loss Eq. (3), we sample two independent sets of noise and timesteps: $(\epsilon_{\text{FM}}, t_{\text{FM}})$ for FM loss, and $(\epsilon_{\text{MF}}, t_{\text{MF}}, r_{\text{MF}})$ for MF loss. For each data sample $\mathbf{x}_0 \sim p_{\text{data}}$, we compute the total loss as the sum of FM and MF losses.[2] We remark that while reusing the same $(\epsilon, t)$ for both FM and MF objectives is possible, it often destabilizes training. The procedure is summarized in Algorithm 1.

**Flow matching warm-up.** Training flow maps is typically more expensive than training flow models due to additional forward passes. For example, computing the MF loss requires Jacobian–vector product (JVP) computations, which can cause memory issues if attention operations are not carefully optimized (Lu & Song, 2025). The cost further increases when model guidance is applied, since guidance-aware targets must be computed.

To mitigate this, we adopt a two-stage strategy: first train a flow model with FM loss, then convert it into DMF and fine-tune with MF loss added. We find that pretrained flow models adapt rapidly to flow maps, especially when their representations are strong, *e.g.*, models trained longer or enhanced with representation alignment (Yu et al., 2025) converge faster (see Tab. 1). As a result, our strategy achieves significantly better scaling than training a flow map from scratch (see Fig. 4).

**Adaptive weighted Cauchy loss.** In practice, the MF loss exhibits high variance, which can hinder stable training. Prior works (Song & Dhariwal, 2024; Geng et al., 2025b; Lu & Song, 2025; Geng et al., 2025a) introduce robust losses to address this. Inspired by these, we employ the Cauchy (Lorentzian) loss (Black & Anandan, 1996; Barron, 2019), defined as

$$\mathcal{L}_{\text{Cauchy}}(\theta) = \log\big(\mathcal{L}_{\text{MF}}(\theta) + c\big), \tag{5}$$

where $c > 0$ is a constant. Like Huber (Song & Dhariwal, 2024) and $\ell_1$ losses (Geng et al., 2025b), the Cauchy loss behaves linearly near zero but suppresses the effect of large outliers.

To further improve stability, we follow Karras et al. (2024b) and incorporate adaptive weighting by modeling the residual error distribution of the MSE as Cauchy for each $(t, r)$ pair. This yields the

---

[2]In contrast, Geng et al. (2025b) splits batch into two groups (*e.g.*, 75% for FM and 25% for MF).

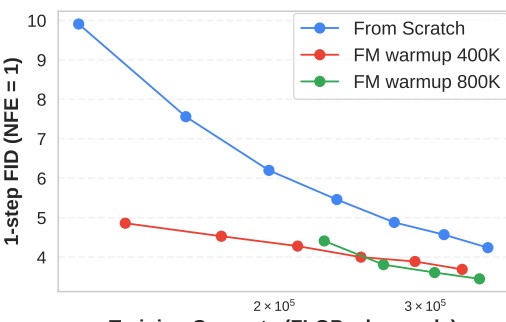

Figure 4: **Effect of Flow Matching warmup.** We plot 1-step FID for DMF-L/2 trained from scratch, DMF-L/2 fine-tuned from SiT-L/2 400K and 800K pretrained models. We plot total training compute used for training. We see that fine-tuned model quickly recovers 1-step performance, and DMF-L/2 fine-tuned from 800K SiT-L/2 model achieves better performance than others, while using fewer total training flops.

adaptive weighted Cauchy loss (see Appendix A.1 for derivation):

$$\mathcal{L}_{\text{DMF}}(\theta) = \mathbb{E}_{\mathbf{x}_t, r}\left[\log\left(e^{-\phi(t,r)}\left\|\mathbf{u}_\theta(\mathbf{x}_t, t, r) - \mathbf{v}^{\text{tgt}}(\mathbf{x}_t, t) - (r-t)\frac{\mathrm{d}\mathbf{u}_\theta}{\mathrm{d}t}\right\|^2 + 1\right) + \frac{\phi(t,r)}{2}\right], \quad (6)$$

where $\phi : [0,1] \times [0,1] \to \mathbb{R}$ is a weighting function. The same formulation is used for FM loss.

**Time proposal.** Sampling timesteps from a logit-normal distribution has proven effective for training flow models (Karras et al., 2022; Esser et al., 2024). For flow maps, however, we require $(t, r)$ pairs with $t > r$. Thus, we follow Geng et al. (2025b) to sampler $(t, r)$ by drawing two logit-normal samples and taking their maximum and minimum.

Notably, as shown in Fig. 3, DMF models converted from flow models already predict accurate average velocities when $r$ is close to $t$. For 1-step generation, it is therefore beneficial to sample pairs where $t$ and $r$ are far apart, particularly with $r$ close to zero. To encourage this, we modify the proposal distribution accordingly, which accelerates 1-step generative modeling (see Appendix A.2).

**Samplers.** Note that the generation for DMF model can be done by Euler's method, *i.e.*, $\mathbf{x}_r = \mathbf{x}_t + (r-t)\mathbf{u}_\theta(\mathbf{x}_t, t, r)$. Alternatively, as the model becomes capable of 1-step sampling, we also consider restart sampler (*i.e.*, used in consistency models (Song et al., 2023)), which predicts $\hat{\mathbf{x}}_0$, and diffuses into $\mathbf{x}_r$. Formally, the algorithm of restart sampler can be written as follows:

$$\hat{\mathbf{x}}_0 \leftarrow \mathbf{x}_t - t\mathbf{u}_\theta(\mathbf{x}_t, t, 0), \qquad \mathbf{x}_r \leftarrow \alpha_r\hat{\mathbf{x}}_0 + \sigma_r\epsilon', \quad \text{where} \ \ \epsilon' \sim \mathcal{N}(\mathbf{0}, \mathbf{I}) \qquad (7)$$

Note that restart sampler is as performative as Euler sampler, and we observe trade-offs between them when evaluated with different metrics, *e.g.*, see Fig. 5. We also examine stochastic samplers that interplays between Euler and restart sampler (Kim et al., 2024) in Appendix C.

## 4 EXPERIMENTS

**Dataset and model.** We conduct our experiments on ImageNet (Deng et al., 2009) dataset following ADM (Dhariwal & Nichol, 2021) preprocessing protocols. We use latent Diffusion Transformer (DiT; Peebles & Xie 2023) as our backbone, where we perform generative modeling on the latent space using pretrained VAE (Rombach et al., 2022). To train flow models, we follow SiT (Ma et al., 2024) and REPA (Yu et al., 2025) when using representation alignment. See Appendix B for details.

**Implementation.** For each experiment, we use BF16 mixed-precision to prevent overflow, and Flash-Attention (Dao, 2024; Shah et al., 2024) to save GPU memory and accelerate training. Similar to (Lu & Song, 2025), we use customize the Flash-Attention kernel to support JVP computation.

**Evaluation.** We evaluate the 1-step (NFE = 1) and 2-step (NFE = 2) generation with Euler solver by default. For comparison, we use Fréchet Inception Distance (FID; Heusel et al. 2017), Inception score (IS; Salimans et al. 2016), and Fréchet Distance with DINOv2-L/14 (Oquab et al., 2023) (FD$_{\text{DINOv2}}$; Stein et al. 2023) evaluated on 50K samples. See Appendix D for details.

### 4.1 ABLATION STUDY

We validate the effectiveness of DMF and study the effect of each component through experiments. Specifically, we aim to answer the following questions:

Table 1: **Ablation study.** All models are fine-tuned from flow model (SiT-L/2) trained for 400K iterations. Depth denotes the number of encoder layers for DMF models, where total number of layers is 24. MG denotes the usage of model guidance (MG) during fine-tuning, and REPA denotes usage of representation alignment during flow model training. We report 1-step (NFE = 1) and 2-step (NFE = 2) FID, IS, FD$_{\text{DINOv2}}$ for each MF-L/2 and DMF-L/2 model. We denote ↓ and ↑ to indicate whether lower or higher values are better, respectively.

| Method | Depth | MG | REPA | 1-step (NFE = 1) | | | 2-step (NFE = 2) | | |
|---|---|---|---|---|---|---|---|---|---|
| | | | | FID ↓ | IS ↑ | FD$_{\text{DINOv2}}$ ↓ | FID ↓ | IS ↑ | FD$_{\text{DINOv2}}$ ↓ |
| MF-L/2 | - | ✗ | ✗ | 20.6 | 72.7 | 540.1 | 18.1 | 77.6 | 476.3 |
| DMF-L/2 | 12 | ✗ | ✗ | 21.9 | 69.3 | 545.1 | 17.9 | 79.2 | 477.9 |
| DMF-L/2 | 16 | ✗ | ✗ | 20.1 | 75.0 | 531.5 | 17.6 | 80.4 | 470.5 |
| DMF-L/2 | 18 | ✗ | ✗ | 19.3 | 79.0 | 531.6 | 17.3 | 81.4 | 461.1 |
| DMF-L/2 | 20 | ✗ | ✗ | 19.5 | 79.2 | 541.4 | 17.4 | 81.1 | 462.8 |
| MF-L/2 | - | ✔ | ✗ | 5.27 | 185.1 | 291.2 | 4.09 | 214.9 | 215.7 |
| DMF-L/2 | 18 | ✔ | ✗ | 4.53 | 197.8 | 275.6 | 3.58 | 225.3 | 197.4 |
| MF-L/2 | - | ✗ | ✔ | 15.8 | 90.5 | 421.7 | 12.6 | 102.7 | 361.2 |
| DMF-L/2 | 18 | ✗ | ✔ | 14.2 | 99.7 | 419.9 | 11.8 | 105.6 | 340.2 |
| MF-L/2 | - | ✔ | ✔ | 3.65 | 219.8 | 205.2 | 2.63 | 257.3 | 136.8 |
| DMF-L/2 | 18 | ✔ | ✔ | 3.10 | 229.8 | 199.7 | 2.51 | 264.6 | 127.3 |

- How effective is DMF architecture in learning flow map? (Tab. 1, Tab. 2)
- How does the representation quality affects the few-step generative modeling? (Tab. 1, Fig. 4)
- How can we efficiently train flow map model? (Fig. 4, Tab. 2)

**Decoder depth ablation.** We analyze the effectiveness of DMF when fine-tuning from flow models. We first train SiT-L/2 model for 400K iterations, then convert into DMF model and fine-tune for 100K iterations. We compare DMF models with MeanFlow(MF;Geng et al. 2025b), where we strictly follow their setup including model guidance scale. In Tab. 1, we report FID, IS, and FD$_{\text{DINOv2}}$ of fine-tuned MF-L/2 and DMF-L/2 models with different depth. We observe that DMF-L/2 achieves lower FID and FD$_{\text{DINOv2}}$, and higher IS compared to MF-L/2 when depth is properly chosen. Notably, we found depth=18 (*i.e.*, 6 blocks for decoder) performs the best, and depth=16 and depth=20 are comparable, which aligns with Fig. 3. When fine-tuned with MG, DMF-L/2 significantly improves 1-step and 2-step performance, and significantly outperforms MF-L/2.

**Effect of encoder representation.** Furthermore, we investigate the effect of representation quality in learning flow map. To this end, we train SiT-L/2+REPA for 400K iterations following (Yu et al., 2025) and fine-tune MF-L/2 and DMF-L/2 model with it. Note that we do not use REPA during fine-tuning as it shows diminishing gain. As shown in Tab. 1, both MF-L/2 and DMF-L/2 achieves better performance when initialized with SiT-L/2+REPA, while DMF-L/2 outperforms MF-L/2. The results are consistent when applied with MG. The results show that the encoder representation helps flow map modeling, where DMF architecture achieves higher gain due to its design.

**Training efficiency.** Next, we examine the efficiency of flow-matching warmup. To this end, we compare DMF-L/2 trained from scratch, and DMF-L/2 initialized from SiT-L/2 trained for 400K and 800K iterations. We use same MG config for each training. In Fig. 4, we plot the 1-step FID (NFE = 1) and training FLOPS for each approach. We observe that the flow models quickly adapts to flow maps, and starting from SiT-L/2 800K model achieves the best performance when using same training compute. We hypothesize that longer flow model training leads to better representation, which helps learning flow map. As of practical consideration, our results show that allocating training budget more on flow model training is more efficient, as adapting to flow map is easier yet expensive.

## 4.2 COMPARISON

**Comparison with MeanFlow.** Given the observations from Sec. 4.1, we conduct system-level comparison between DMF and MF models of various sizes (B/2, L/2, and XL/2) with same guidance configuration. For DMF models, we train with flow matching loss for 160 epochs, and continue train with DMF loss for 40 and 80 epochs. Tab. 2 compares FID-50K of MF and DMF models. We observe

Table 2: **Comparison with MeanFlow** on ImageNet 256×256. We compare FID of DMF and MeanFlow (MF; Geng et al. 2025a) models of various size. For each, we apply same guidance as of MF models. FID results of MF models are excerpted from their paper. DMF models do flow matching warm-up for 160 epochs, and fine-tuned for 40 and 80 epochs, i.e., 200 and 240 epochs total, respectively. Note that DMF models require fewer training FLOPs than MF.

| Model | Epochs | NFE | #Params | FID ↓ |
|---|---|---|---|---|
| MF-B/2 | 240 | 1 | 130M | 6.17 |
| **DMF-B/2** | 200 | 1 | 130M | 6.08 |
| | 240 | 1 | 130M | **5.63** |
| MF-L/2 | 240 | 1 | 459M | 3.84 |
| **DMF-L/2** | 200 | 1 | 459M | 3.45 |
| | 240 | 1 | 459M | **3.24** |
| MF-XL/2 | 240 | 1 | 676M | 3.43 |
| **DMF-XL/2** | 200 | 1 | 675M | 3.02 |
| | 240 | 1 | 675M | **2.83** |
| MF-XL/2 | 240 | 2 | 676M | 2.93 |
| **DMF-XL/2** | 240 | 2 | 675M | **2.56** |

Table 3: **ImageNet 256×256 benchmark.** 2× denotes usage of CFG and † denotes usage of guidance interval (Kynkäänniemi et al., 2024).

| Method | NFE | #Params | FID ↓ |
|---|---|---|---|
| **GANs / Normalizing Flows / Autoregressive models** | | | |
| StyleGAN-XL (Sauer et al., 2022) | 1 | 166M | 2.30 |
| VAR-d30 (Tian et al., 2024) | 2×10 | 2B | 1.92 |
| MAR-H/2 (Li et al., 2024) | 2×256 | 943M | 1.55 |
| STARFlow (Gu et al., 2025) | 1 | 1.4B | 2.40 |
| **Diffusion / Flow models** | | | |
| ADM (Dhariwal & Nichol, 2021) | 2×250 | 554M | 10.94 |
| LDM (Rombach et al., 2022) | 2×250 | 400M | 3.60 |
| RIN (Jabri et al., 2023) | 1000 | 410M | 3.42 |
| SimDiff (Hoogeboom et al., 2023) | 2×512 | 2B | 2.77 |
| U-ViT-H/2 (Bao et al., 2023) | 2×50 | 501M | 2.29 |
| DiffIT (Hatamizadeh et al., 2024) | 2×250 | 561M | 1.73 |
| DiT-XL/2 (Peebles & Xie, 2023) | 2×250 | 675M | 2.27 |
| SiT-XL/2 (Ma et al., 2024) | 2×250 | 675M | 2.06 |
| SiT-XL/2+REPA† (Yu et al., 2025) | 434 | 675M | **1.37** |
| **Few-step diffusion / flow models** | | | |
| Shortcut-XL/2 (Frans et al., 2025) | 1 | 676M | 10.60 |
| | 4 | 676M | 7.80 |
| IMM-XL/2 (Zhou et al., 2025) | 2×1 | 676M | 7.77 |
| | 2×2 | 676M | 3.99 |
| | 2×4 | 676M | 2.51 |
| MF-XL/2+ (Geng et al., 2025b) | 2 | 676M | 2.20 |
| **DMF-XL/2+ (ours)** | 1 | 675M | **2.16** |
| | 2 | 675M | **1.64** |
| | 4 | 675M | **1.51** |

Table 4: **ImageNet 512×512 benchmark.** 2× denotes usage of CFG, † denotes usage of guidance interval (Kynkäänniemi et al., 2024), and * denotes usage of AutoGuidance (Karras et al., 2024a).

| Method | NFE | #Params | FID ↓ |
|---|---|---|---|
| **GANs / Normalizing Flows / Autoregressive models** | | | |
| StyleGAN-XL (Sauer et al., 2022) | 1 | 168M | 2.41 |
| STARFlow (Gu et al., 2025) | 1 | 3B | 3.00 |
| VAR-d36 (Tian et al., 2024) | 2×10 | 2.3B | 2.63 |
| MAR-L (Li et al., 2024) | 2×64 | 481M | 1.73 |
| **Diffusion / Flow models** | | | |
| ADM (Dhariwal & Nichol, 2021) | 2×250 | 559M | 3.85 |
| SimDiff (Hoogeboom et al., 2023) | 2×512 | 2B | 3.02 |
| DiffIT (Hatamizadeh et al., 2024) | 2×250 | 561M | 2.67 |
| DiT-XL/2 (Peebles & Xie, 2023) | 2×250 | 675M | 3.04 |
| SiT-XL/2 (Ma et al., 2024) | 2×250 | 675M | 2.62 |
| SiT-XL/2+REPA† (Yu et al., 2025) | 460 | 675M | 1.37 |
| EDM2-XXL (Karras et al., 2024b) | 2×63 | 1.5B | 1.81 |
| EDM2-XXL† (Kynkäänniemi et al., 2024) | 82 | 1.5B | 1.40 |
| EDM2-XXL* (Karras et al., 2024a) | 2×63 | 1.5B | **1.25** |

| Method | NFE | #Params | FID ↓ |
|---|---|---|---|
| **Few-step Diffusion / Flow models** | | | |
| sCD-L (Lu & Song, 2025) | 1 | 778M | 2.55 |
| | 2 | 778M | 2.04 |
| sCD-XL (Lu & Song, 2025) | 1 | 1.1B | 2.40 |
| | 2 | 1.1B | 1.93 |
| sCD-XXL (Lu & Song, 2025) | 1 | 1.5B | 2.28 |
| | 2 | 1.5B | 1.88 |
| AYF-S* (Sabour et al., 2025) | 1 | 280M | 3.32 |
| | 2 | 280M | 1.87 |
| | 4 | 280M | 1.70 |
| **DMF-XL/2+ (ours)** | 1 | 675M | **2.12** |
| | 2 | 675M | **1.75** |
| | 4 | 675M | **1.68** |

that DMF models trained for 200 epochs achieves lower FID scores than MF models trained for 240 epochs, demonstrating its efficiency in training. Furthermore, DMF models trained for 240 epochs significantly outperforms MF models. In particular, DMF-XL/2 achieves 1-step FID=3.10, achieving state-of-the-art result on 1-step diffusion / flow models. We also remark that the training FLOPs of each DMF is smaller than MF counterpart, as flow matching is much cheaper than flow map training.

**ImageNet benchmark.** Following our observation in Tab. 1, we initialize DMF model from longer trained SiT-XL/2+REPA to explore the limit of DMF model, and conduct fine-tuning with model guidance applied. Then we compare our final model, DMF-XL/2+, with other few-step models (Frans et al., 2025; Zhou et al., 2025; Geng et al., 2025b; Lu & Song, 2025; Sabour et al., 2025), diffusion / flow models (Dhariwal & Nichol, 2021; Rombach et al., 2022; Jabri et al., 2023; Bao et al., 2023; Hatamizadeh et al., 2024; Peebles & Xie, 2023; Ma et al., 2024; Yu et al., 2025; Karras et al., 2024b), and various generative models such as GANs (Sauer et al., 2022), Normalizing Flows (Gu et al., 2025), and autoregressive models (Tian et al., 2024; Li et al., 2024).

**ImageNet 256×256 comparison.** In Tab. 3, we notice that DMF-XL/2+ achieves 1-NFE FID=2.16, significantly outperforms 1-step diffusion / flow baselines, and strong 1-NFE baselines StyleGAN-XL (FID=2.30) and STARFlow (FID=2.40). Furthermore, DMF-XL/2+ achieves 4-NFE FID of 1.51, matching performance of SiT-XL/2+REPA (FID=1.37), while using ×100 less computation.

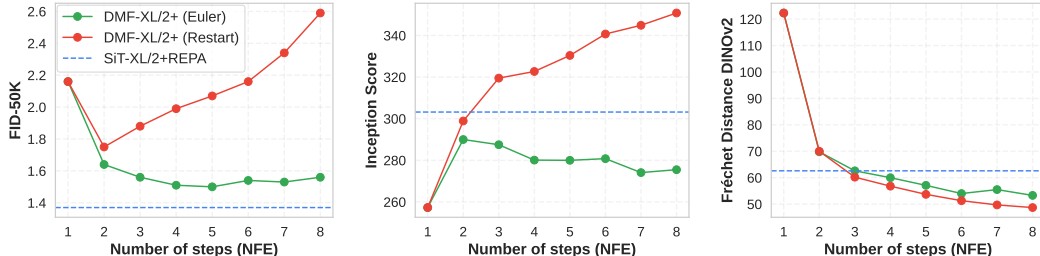

Figure 5: **Euler vs. Restart samplers.** We compare Euler and Restart samplers with DMF-XL/2+ trained on ImageNet 256×256. FID-50K, Inception score (IS), and Fréchet distance DINOv2 ($FD_{DINOv2}$) are reported. We plot results for SiT-XL/2+REPA with CFG.

**ImageNet 512×512 comparison.** We further experiment DMF on ImageNet 512 resolution. Similarly, we train SiT-XL/2+REPA for 400 epochs, and conduct DMF fine-tuning for 140 epochs. Note that we observe training instability during DMF fine-tuning on higher resolution. Thus, we add QK normalization (Dehghani et al., 2023) with RMSNorm (Zhang & Sennrich, 2019) to enhance training stability and image generation fidelity. We refer to Appendix E for details. As shown in Tab. 4, DMF-XL/2+ achieves 1-NFE FID=2.12 and 2-NFE FID=1.75, outperforming the prior art sCD-XXL (Lu & Song, 2025). Furthermore, DMF-XL/2+ achieves 4-NFE FID=1.68, outperforming AYF-S, which uses AutoGuidance (Karras et al., 2024a) during distillation to enhance generation quality. The qualitative samples are in Fig. 2 and Appendix H.

**Choice of sampler.** Lastly, we compare the Euler and restart samplers on various metrics (FID, IS, and $FD_{DINOv2}$). For reference, we report SiT-XL/2+REPA results using Heun's method with 128 steps, and guidance interval applied (see Appendix E). In Fig. 5, we plot the results. We observe that DMF-XL/2+ with Euler sampler achieves lower $FD_{DINOv2}$ than SiT-XL/2+REPA when using NFE larger than 3, while showing slightly higher FID. Furthermore, when with restart sampler, DMF-XL/2+ achieves higher IS and lower $FD_{DINOv2}$. As noticed in quantitative metrics, we find that restart solver generates more diverse samples due to the stochasticity throughout sampling, which we qualitatively visualize in Appendix H. We notice that our model is capable of applying diverse samplers, where there is a trade-off in terms of different metrics, *e.g.*, Euler sampler shows better FID, while restart sampler better scales in terms of IS and $FD_{DINOv2}$.

## 5 RELATED WORK

**1-step diffusion and flow models.** Research on accelerating diffusion models has advanced from both practical and theoretical perspectives. To mitigate the high sampling cost of pretrained diffusion models, distillation-based methods have shown strong promise (Salimans & Ho, 2022; Meng et al., 2023; Yin et al., 2024b;a; Heek et al., 2024). Beyond distillation, *consistency models* (Song et al., 2023; Song & Dhariwal, 2024; Lu & Song, 2025; Geng et al., 2025b; Zhou et al., 2025) introduced a principled approach to learn a single-step denoiser by enforcing consistency constraints across adjacent timesteps. More recently, several works have proposed learning *flow maps* (Kim et al., 2024; Boffi et al., 2024; 2025; Geng et al., 2025a; Sabour et al., 2025), which generalize consistency models by modeling the transition between arbitrary pairs of timesteps. While prior efforts primarily focus on designing objectives for learning flow maps, our work instead investigates architectural design, drawing on the structural similarities between flow models and flow maps.

**Decoupled architectures for generative modeling.** Another line of work explores *decoupled architectures*, typically separating encoder and decoder roles, to improve visual generative modeling. Several studies (Yu et al., 2025; Wang & He, 2025; Wang et al., 2025) demonstrate that strengthening the representational capacity of the encoder in diffusion transformers (Peebles & Xie, 2023) enhances both scalability and performance of diffusion and flow models. Alternatively, MAR (Li et al., 2024) employs an encoder–decoder design for masked autoregressive generation, inspired by the success of MAE (He et al., 2022) in representation learning. Our work shares this emphasis on the role of representation, but extends the perspective toward flow map learning, enabling few-step generation while maintaining alignment with underlying flow models.

## 6 CONCLUSION

This paper introduces Decoupled MeanFlow (DMF), an efficient and effective method to learn flow maps for fast generative modeling. Specifically, we show that DMF can seamlessly convert flow models into flow maps, and fine-tuning DMF models with flow map training loss achieves high-quality 1-step generative models. We hope our work promotes future research about efficient inference-time scaling, as well as post-training algorithms that can enhance the generation quality and controllability. We discuss limitations and future directions in Appendix G.

## ETHICS STATEMENT

Our work advances visual generative AI, which carries potential risks of misuse such as disinformation, deepfakes, or biased outputs. We emphasize that our methods are intended for responsible applications, and we encourage safeguards like watermarking, dataset auditing, and alignment techniques to ensure safe and ethical deployment.

## ACKNOWLEDGEMENT

This work was partly supported by Institute of Information & Communications Technology Planning & Evaluation (IITP) grant funded by the Korea government (MSIT) (No. RS-2019-II190075, Artificial Intelligence Graduate School Program (KAIST); No. RS-2024-00509279, Global AI Frontier Lab). We thank Kihyuk Sohn for valuable discussions and insightful feedback throughout the project.

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

## A  TRAINING ALGORITHM

We provide a detailed algorithm to train Decoupled MeanFlow model in Algorithm 1.

---

**Algorithm 1** Training Algorithm for Decoupled MeanFlow

---

**Require:** Dataset $\mathcal{D}$, flow map $\mathbf{u}_\theta$, weighting function $\phi$, learning rate $\eta > 0$, constant $c > 0$, time proposal for FM loss $(\mu_{\text{FM}}, \Sigma_{\text{FM}})$, time proposal for MF loss $(\mu_{\text{MF}}^{(1)}, \mu_{\text{MF}}^{(2)}, \Sigma_{\text{MF}}^{(1)}, \Sigma_{\text{MF}}^{(2)})$, model guidance scale $\omega > 0$, class dropout probability $q > 0$, guidance interval $[g_{\text{low}}, g_{\text{high}}]$

1: **while** not converged **do**
2:    Sample $(\mathbf{x}, \mathbf{y}) \sim \mathcal{D}$ and drop $\mathbf{y}$ with probability $q$
3:    Sample $\tau_{\text{FM}} \sim \mathcal{N}(\mu_{\text{FM}}, \Sigma_{\text{FM}})$, $t_{\text{FM}} \leftarrow (1 + e^{-\tau_{\text{FM}}})^{-1}$, and noise $\epsilon_{\text{FM}} \sim \mathcal{N}(\mathbf{0}, \mathbf{I})$
4:    Diffuse $\mathbf{x}_{t_{\text{FM}}} \leftarrow \alpha_{t_{\text{FM}}} \mathbf{x}_0 + \sigma_{t_{\text{FM}}} \epsilon_{\text{FM}}$, and $\mathbf{v}_{t_{\text{FM}}} \leftarrow \alpha'_{t_{\text{FM}}} \mathbf{x}_0 + \sigma'_{t_{\text{FM}}} \epsilon_{\text{FM}}$
5:    $\mathbf{v}_{\text{FM}}^{\text{tgt}} \leftarrow \mathbf{v}_{t_{\text{FM}}} + \omega(\mathbf{u}_\theta(\mathbf{x}_{t_{\text{FM}}}, t_{\text{FM}}, t_{\text{FM}}, \mathbf{y}) - \mathbf{u}_\theta(\mathbf{x}_{t_{\text{FM}}}, t_{\text{FM}}, t_{\text{FM}})$ if $t_{\text{FM}} \in [g_{\text{low}}, g_{\text{high}}]$ else $\mathbf{v}_{t_{\text{FM}}}$
6:    Detach gradient $\mathbf{v}_{\text{FM}}^{\text{tgt}} \leftarrow \texttt{sg}(\mathbf{v}_{\text{FM}}^{\text{tgt}})$
7:    $\mathcal{L}_{\text{FM}}(\theta) \leftarrow \log\left(e^{-\phi(t_{\text{FM}}, t_{\text{FM}})} \left\|\mathbf{u}_\theta(\mathbf{x}_{t_{\text{FM}}}, t_{\text{FM}}, t_{\text{FM}}) - \mathbf{v}_{\text{FM}}^{\text{tgt}}\right\|^2 + c\right) + \phi(t_{\text{FM}}, t_{\text{FM}})$
8:    Sample $\tau_1 \sim \mathcal{N}\left(\mu_{\text{MF}}^{(1)}, \Sigma_{\text{MF}}^{(1)}\right), \tau_2 \sim \mathcal{N}\left(\mu_{\text{MF}}^{(2)}, \Sigma_{\text{MF}}^{(2)}\right), t_1 \leftarrow (1 + e^{-\tau_1})^{-1}, t_2 \leftarrow (1 + e^{-\tau_2})^{-1}$
9:    Let $t_{\text{MF}}, r_{\text{MF}} \leftarrow \max(t_1, t_2), \min(t_1, t_2)$, noise $\epsilon_{\text{MF}} \sim \mathcal{N}(\mathbf{0}, \mathbf{I})$
10:   Diffuse $\mathbf{x}_{t_{\text{MF}}} \leftarrow \alpha_{t_{\text{MF}}} \mathbf{x}_0 + \sigma_{t_{\text{MF}}} \epsilon_{\text{MF}}$, and $\mathbf{v}_{t_{\text{MF}}} \leftarrow \alpha'_{t_{\text{MF}}} \mathbf{x}_0 + \sigma'_{t_{\text{MF}}} \epsilon_{\text{MF}}$
11:   $\mathbf{v}_{\text{MF}}^{\text{tgt}} \leftarrow \mathbf{v}_{t_{\text{MF}}} + \omega(\mathbf{u}_\theta(\mathbf{x}_{t_{\text{MF}}}, t_{\text{MF}}, t_{\text{MF}}, \mathbf{y}) - \mathbf{u}_\theta(\mathbf{x}_{t_{\text{MF}}}, t_{\text{MF}}, t_{\text{MF}})$ if $t_{\text{MF}} \in [g_{\text{low}}, g_{\text{high}}]$ else $\mathbf{v}_{t_{\text{MF}}}$
12:   Set target $\mathbf{u}^{\text{tgt}} = \mathbf{v}_{\text{MF}}^{\text{tgt}} + (r_{\text{MF}} - t_{\text{MF}})\frac{\mathrm{d}\mathbf{u}_\theta}{\mathrm{d}t}$ and detach gradient $\mathbf{u}_{\text{MF}}^{\text{tgt}} \leftarrow \texttt{sg}(\mathbf{u}_{\text{MF}}^{\text{tgt}})$
13:   $\mathcal{L}_{\text{MF}}(\theta) \leftarrow \log\left(e^{-\phi(t_{\text{MF}}, r_{\text{MF}})} \left\|\mathbf{u}_\theta(\mathbf{x}_{t_{\text{MF}}}, t_{\text{MF}}, r_{\text{MF}}) - \mathbf{u}_{\text{MF}}^{\text{tgt}}\right\|^2 + c\right) + \phi(t_{\text{MF}}, r_{\text{MF}})$
14:   Compute total loss $\mathcal{L}(\theta) \leftarrow \mathcal{L}_{\text{FM}}(\theta) + \mathcal{L}_{\text{MF}}(\theta)$
15:   Update $\theta \leftarrow \theta - \eta\nabla_\theta\mathcal{L}(\theta), \phi \leftarrow \phi - \eta\nabla_\phi\mathcal{L}(\theta)$
16: **end while**

---

### A.1  TRAINING LOSS

The loss weighting is shown to be important when training diffusion and flow models. As such, Karras et al. (2020) proposed to adaptively learn the weighting function by casting as a multi-task learning problem. In particular, let us denote $\mathcal{L}_t$ be the flow matching loss for timestep $t$. Following Kendall et al. (2018), they consider each loss $\mathcal{L}_t$ as a Gaussian distribution modeled standard deviation $\sigma_t$, where the maximum likelihood estimation of overall loss results in

$$\mathcal{L} = \frac{1}{2}\mathbb{E}_t\left[\frac{1}{\sigma_t^2}\mathcal{L}_t + \log\sigma_t^2\right] = \frac{1}{2}\mathbb{E}_t\left[\frac{\mathcal{L}_t}{e^{u_t}} + u_t\right], \tag{8}$$

where $u_t = \log\sigma_t^2$ is a log-variance. At a high-level, if the model is uncertain about the task at time $t$, *i.e.*, if $u_t$ is high, then the contribution of $\mathcal{L}_t$ is decreased. In practice, they use an MLP layer with Fourier time encoding to model $u_t = \phi(t)$.

Similarly, we consider the loss for flow map training as $\mathcal{L}_{t,r}$. However, as the flow map loss $\mathcal{L}_{t,r}$ is intrinsically of high-variance, and prone to the outliers, modeling with Gaussian distribution may be suboptimal. To this end, we consider Cauchy distribution, has heavier tails than Gaussian. Note that the probability density function of Cauchy distribution is given as

$$p(x; x_0, \gamma) = \frac{1}{\pi\gamma}\frac{1}{1 + (\frac{x-x_0}{\gamma})^2},$$

where $x_0$ is a location and $\gamma$ is a scale parameter. Then we model each loss output with additional parameter $\gamma_{t,r}$, and the maximum likelihood estimation of the overall loss is given by

$$\mathcal{L} = \mathbb{E}_{t,r}\left[\log\left(\frac{1}{\gamma_{t,r}^2}\mathcal{L}_{t,r} + 1\right) + \frac{1}{2}\log\gamma_{t,r}^2\right] = \mathbb{E}_{t,r}\left[\log\left(\frac{\mathcal{L}_{t,r}}{e^{u_{t,r}}} + 1\right) + \frac{1}{2}u_{t,r}\right], \tag{9}$$

where $u_{t,r} = \log\gamma_{t,r}^2$ and we omit terms for $\pi$ as it does not affect training. For implementation, we concatenate the positional embeddings of $t$ and $r$ and use an MLP layer to train $u_{t,r} = \phi(t, r)$, which gives us Eq. (6).

## A.2 TIMESTEP PROPOSAL

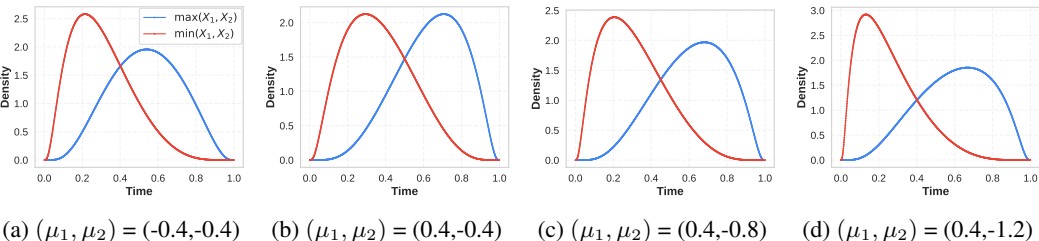

(a) $(\mu_1, \mu_2)$ = (-0.4,-0.4)  (b) $(\mu_1, \mu_2)$ = (0.4,-0.4)  (c) $(\mu_1, \mu_2)$ = (0.4,-0.8)  (d) $(\mu_1, \mu_2)$ = (0.4,-1.2)

Figure 6: **Time proposal.** The probability density distribution of $\max(X_1, X_2)$ and $\min(X_1, X_2)$, where $X_1 \sim \text{LogitNormal}(\mu_1, 1, 0)$, and $X_2 \sim \text{LogitNormal}(\mu_2, 1.0)$. The red line depicts distribution of $r_{\text{MF}}$ and blue line depicts distribution of $t_{\text{MF}}$. We use (d) as our default choice.

As we mentioned in Sec. 3, we sample $t$ and $r$ from the maximum and minimum of two logit-normal distributions. To characterize the distribution, note that for any two independent continuous random variables $X, Y$ with densities $f_X, f_Y$, and CDFs $F_X, F_Y$, the order statistics give us following:

$$F_{\max}(z) = \Pr[\max(X, Y) \leq z] = F_X(z)F_Y(z),$$

$$f_{\max}(z) = \frac{\mathrm{d}}{\mathrm{d}z} F_X(z)F_Y(z) = f_X(z)F_Y(z) + F_X(z)f_Y(z),$$

$$F_{\min}(z) = \Pr[\min(X, Y) \leq z] = 1 - \Pr[X > z, Y > z] = 1 - (1 - F_X(z))(1 - F_Y(z)),$$

$$f_{\min}(z) = \frac{\mathrm{d}}{\mathrm{d}z}[F_X(z) + F_Y(z) - F_X(z)F_Y(z)] = f_X(z)[1 - F_Y(z)] + f_Y(z)[1 - F_X(z)],$$

where $f_{\max}, f_{\min}$ are densities and $F_{\max}, F_{\min}$ are CDFs of $\max(X, Y)$ and $\min(X, Y)$, respectively. We consider logit-normal distribution with location of $\mu$ and scale of 1, *i.e.*, $\text{LN}(\mu, 1)$. Then for two independent logit-normal distributions $X_1$ and $X_2$ with scale parameters $\mu_1$ and $\mu_2$, the densities of $\max(X_1, X_2)$ and $\min(X_1, X_2)$ is given as follows:

$$f_{\max}(z) = \frac{\phi(\text{logit}(z) - \mu_1)\Phi(\text{logit}(z) - \mu_2) + \phi(\text{logit}(z) - \mu_2)\Phi(\text{logit}(z) - \mu_1)}{z(1 - z)}$$

$$f_{\min}(z) = \frac{\phi(\text{logit}(z) - \mu_1)[1 - \Phi(\text{logit}(z) - \mu_2)] + \phi(\text{logit}(z) - \mu_2)[1 - \Phi(\text{logit}(z) - \mu_1)]}{z(1 - z)},$$

where $\text{logit}(z) = \log \frac{z}{1-z}$, $\phi(z) = \frac{1}{\sqrt{2\pi}}e^{-z^2/2}$, $\Phi(z) = \int_{-\infty}^{z} \phi(u)\mathrm{d}u$. By using the above formula, we plot the distribution of $\max(X_1, X_2)$ and $\min(X_1, X_2)$ in Fig. 6 by varying $\mu_1$ and $\mu_2$. Note that as we increase the gap between $\mu_1$ and $\mu_2$, the $\min(X_1, X_2)$ is sampled close to zero.

We hypothesize that choosing $r$ close to zero improves the 1-step generative modeling. To validate this, we conduct an ablation study with identical setup as in Tab. 1, by varying $\mu_1$ and $\mu_2$. Note that $(\mu_1, \mu_2)$ = (-0.4,-0.4) is the original setup used in MeanFlow (Geng et al., 2025b). As shown in Tab. 5, choosing $r$ closer to zero leads to better 1-step performance. By choosing $(\mu_1, \mu_2)$=(0.4,-1.2), we achieve the best results, which we use as our default configuration.

Table 5: 1-step results by varying $(\mu_1, \mu_2)$.

| $(\mu_1, \mu_2)$ | FID | IS | $\text{FD}_{\text{DINOv2}}$ |
|---|---|---|---|
| (-0.4, -0.4) | 20.3 | 78.1 | 571.4 |
| (0.4, -0.4) | 19.7 | 78.8 | 548.2 |
| (0.4, -0.8) | 19.6 | 78.1 | 540.3 |
| (0.4, -1.2) | 19.3 | 79.0 | 531.6 |

# B IMPLEMENTATION

Table 6: Configurations for ImageNet experiments.

|  | DMF-B/2 | DMF-L/2 | DMF-XL/2 | DMF-XL/2+ | DMF-XL/2+ |
|---|---|---|---|---|---|
| *Model* | | | | | |
| Resolution | 256×256 | 256×256 | 256×256 | 256×256 | 512×512 |
| Params (M) | 130 | 458 | 675 | 675 | 675 |
| FLOPS (G) | 23.1 | 80.7 | 118.6 | 118.6 | 524.6 |
| Hidden dim. | 768 | 1024 | 1152 | 1152 | 1152 |
| Heads | 12 | 16 | 16 | 16 | 16 |
| Patch size | 2×2 | 2×2 | 2×2 | 2×2 | 2×2 |
| Sequence length | 256 | 256 | 256 | 256 | 1024 |
| Layers | 12 | 24 | 28 | 28 | 28 |
| DMF depth | 8 | 18 | 20 | 20 | 20 |
| *Optimization* | | | | | |
| Optimizer | AdamW (Kingma & Ba, 2014; Loshchilov & Hutter, 2017) | | | | |
| Batch size | 256 | | | | |
| Learning rate | 1e-4 | | | | |
| Adam $(\beta_1, \beta_2)$ | (0.9, 0.95) | | | | |
| Adam $\epsilon$ | 1e-8 | | | | |
| Adam weight decay | 0.0 | | | | |
| EMA decay rate | 0.9999 | | | | |
| *Flow model training* | | | | | |
| Training iteration | 800K | 800K | 800K | 4M | 2M |
| Epochs | 160 | 160 | 160 | 800 | 400 |
| Class dropout probability | 0.1 | 0.1 | 0.1 | 0.1 | 0.1 |
| Time proposal $\mu_{\text{FM}}$ | 0.0 | 0.0 | 0.0 | - | 0.0 |
| REPA alignment depth | - | - | - | 8 | 8 |
| REPA vision ecoder | - | - | - | DINOv2-B/14 | DINOv2-B/14 |
| QK-norm | ✗ | ✗ | ✗ | ✗ | ✗ |
| *DMF flow map training* | | | | | |
| Training iteration | 400K | 400K | 400K | 400K | 700K |
| Epochs | 80 | 80 | 80 | 80 | 140 |
| Class dropout probability | 0.1 | | | | |
| Time proposal $\mu_{\text{FM}}$ | 0.0 | | | | |
| Time proposal $(\mu_{\text{MF}}^{(1)}, \mu_{\text{MF}}^{(2)})$ | (0.4, -1.2) | | | | |
| Model guidance scale $\omega$ | 0.5 | 0.6 | 0.6 | 0.6 | 0.6 |
| Guidance interval | [0.0, 1.0] | [0.0, 0.7] | [0.0, 0.7] | [0.0, 0.7] | [0.0, 0.8] |
| QK norm | ✗ | ✗ | ✗ | ✗ | ✔ |

The detailed configurations are in Tab. 6. We use latent Diffusion Transformer (DiT; Peebles & Xie 2023) as our backbone. We use pretrained Stable Diffusion VAE (Rombach et al., 2022) to compress an image with downsampling ratio of 8 and channel dimension of 4, *e.g.*, a 256×256 image is compressed into a latent with size 4×32×32. When training flow models with REPA (Yu et al., 2025), the loss function is given by $\mathcal{L} = \mathcal{L}_{\text{FM}} + \lambda\mathcal{L}_{\text{REPA}}$, where $\mathcal{L}_{\text{REPA}}$ is a cosine-similarity loss between embeddings of intermediate output of transformer layer and vision encoder outputs. We use $\lambda = 0.5$ with 3-layers MLP with SiLU activation (Elfwing et al., 2018) for alignment loss following the original implementation. The evaluation of flow models are in Tab. 7.

To expedite training, we use Flash-Attention v2 (Dao, 2024) or Flash-Attention v3 (Shah et al., 2024). As Flash-Attention kernel do not support JVP computation in pytorch, we follow the approach introduced in Lu & Song (2025). Specifically, we use the same kernels for forward and backward computation as of Flash-Attention, but for JVP computation, we compute the function output and JVP output simultaneously at the forward pass. To this end, we implement the kernel using Triton (Tillet et al., 2019), and merged through `torch.autograd.function`. When compared to non-optimized kernel, *e.g.*, using pytorch's native matrix multiplication to compute JVP, we achieve ×4 GPU memory savings for ImageNet 512×512 model where the sequence length is 1024.

We use automatic mixed-precision with brain floating point 16 (BF16) throughout experiments. We find that using floating point 16 (FP16) often incurs instability, especially when training with model guidance. Furthermore, when training on ImageNet 512×512, we observed large variance of gradient norm during training, which leads to inferior results. To this end, we apply QK normalization (Dehghani et al., 2023) with RMSNorm (Zhang & Sennrich, 2019) during DMF training, which enhances the training stability as well as generation quality.

When training with model guidance (MG), note that the guidance scale $\omega$ is related to the CFG scale. Specifically, note that for FM loss, we enforce the conditional velocity to achieve

$$\mathbf{v}(\mathbf{x}_t, t, \mathbf{y}) = \mathbf{v}_t + \omega(\mathbf{v}(\mathbf{x}_t, t, \mathbf{y}) - \mathbf{v}(\mathbf{x}_t, t)) \Leftrightarrow \mathbf{v}(\mathbf{x}_t, t, \mathbf{y}) = \frac{1}{1 - \omega}(\mathbf{v}_t - \omega \mathbf{v}(\mathbf{x}_t, t)), \quad (10)$$

which becomes equivalent to interpolation of $\mathbf{v}_t$ and unconditional velocity $\mathbf{v}(\mathbf{x}_t, t)$ with guidance scale $\frac{1}{1-\omega}$. For instance, when using $\omega = 0.5$, this becomes guidance scale of 2.0. While original MeanFlow paper uses complicated choice by introducing additional hyperparameter to interpolate between $\mathbf{v}_t$ and CFG velocity, we find applying only MG suffices to achieve good performance. We also tried using distillation (Song & Dhariwal, 2024; Lu & Song, 2025), *i.e.*, using $\mathbf{v}_{\text{tgt}}$ by using the CFG velocity from pretrained flow models, but we did not find performance gain. Alternatively, one can apply Auto-guidance (Karras et al., 2024a), which uses a small and under-fitted model for guidance, during flow map distillation as shown in AYF (Sabour et al., 2025), which we leave as future direction. Lastly, we note that applying guidance is crucial in achieving high-quality 1-step generator; for DMF-XL/2+ model trained without MG, it achieves 1-step FID (NFE = 1) of 8.89 and 4-step FID (NFE = 4) of 7.87. If we apply CFG during inference, 4-step FID (NFE = 8) achieves 2.15, which still lags behind the DMF-XL/2+ with MG (FID=1.51 with NFE = 4).

### B.1 ARCHITECTURE

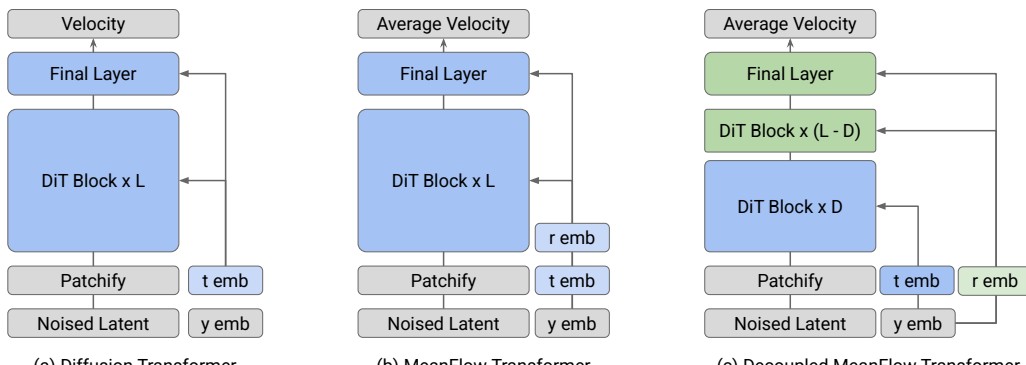

(a) Diffusion Transformer    (b) MeanFlow Transformer    (c) Decoupled MeanFlow Transformer

Figure 7: Comparison of Diffusion Transformer (DiT; Peebles & Xie 2023), MeanFlow DiT (MFT; Geng et al. 2025a), and Decoupled MeanFlow DiT (DMFT; ours).

Fig. 7 depicts the transformer architectures we used. The Diffusion Transformer computes the condition embedding by sum of class embedding ($y$ embed) and timestep embedding ($t$ embed). For MeanFlow Transformer (Geng et al., 2025a; Zhou et al., 2025), an additional timestep embedding layer is used, and the condition embedding is computed by the sum of $y$ embed, $t$ embed, and $r$ embed, where $r$ is the next timestep. Then the condition embedding is fed to all DiT blocks and final layers, *e.g.*, for modulation of outputs from attention layer and feedforward layer with AdaLN (Peebles & Xie, 2023). On the other hand, Decoupled MeanFlow Transformer use same timestep embedding for $r$, and computes condition embedding for encoders by sum of $t$ embed and $y$ embed, and sum of $r$ embed and $y$ embed for decoders. Note that for MeanFlow Transformer, we feed $t - r$ as input to the $r$ embedding layer, following their observation.

## C SAMPLING

We provide details on the sampling algorithms for flow models and flow map models. **Sampling**

**with flow models.** Given the trained velocity prediction model $\mathbf{v}_\theta$, solving the probability flow ODE (Song et al., 2021) with Euler's method is given by

$$\mathbf{x}_t \leftarrow \mathbf{x}_t + (r - t)\mathbf{v}_\theta(\mathbf{x}_t, t), \tag{11}$$

where $t$ is a current timestep and $r$ is a next timestep. To reduce the discretization error and improve the precision, one can use high-order method (Karras et al., 2022; Lu et al., 2022; 2025). For instance, Heun's method is a two-stage algorithm that updates the latents as follows:

$$\hat{\mathbf{x}}_r = \mathbf{x}_t + (r - t)\mathbf{v}_\theta(\mathbf{x}_t, t)$$
$$\mathbf{x}_r = \mathbf{x}_t + \frac{r - t}{2}\big(\mathbf{v}_\theta(\mathbf{x}_t, t) + \mathbf{v}_\theta(\hat{\mathbf{x}}_r, r)\big)$$

Alternatively, one can solve the stochastic differential equation (SDE) with Euler-Maruyama method (Ma et al., 2024). Note that the the SDE is given by

$$\mathrm{d}\mathbf{x}_t = \mathbf{v}_\theta(\mathbf{x}_t, t)\mathrm{d}t - \frac{1}{2}w_t\mathbf{s}_\theta(\mathbf{x}_t, t)\mathrm{d}t + \sqrt{w_t}\,\mathrm{d}\overline{\mathbf{W}}_t, \tag{12}$$

where $\overline{\mathbf{W}}_t$ is a reverse-time Wiener process, and $w_t > 0$ is an arbitrary time-dependent diffusion coefficient, and $\mathbf{s}_\theta$ is an approximation of score function $\mathbf{s}(\mathbf{x}, t) = \nabla \log p_t(\mathbf{x})$. Note that one can compute the score function using the velocity prediction model for $t > 0$ using following formula:

$$\mathbf{s}_\theta(\mathbf{x}_t, t) = \frac{\alpha_t \mathbf{v}_\theta(\mathbf{x}_t, t) - \alpha_t' \mathbf{x}_t}{\sigma_t(\alpha_t'\sigma_t - \alpha_t\sigma_t')},$$

where $\alpha_t, \sigma_t$ are coefficients.

**Sampling with flow maps.** Given the flow maps $u_\theta$, the Euler sampler updates the samples through solving the ODE, *i.e.*,

$$\mathbf{x}_r = \mathbf{x}_t + (r - t)\mathbf{u}_\theta(\mathbf{x}_t, t, r), \tag{13}$$

and the restart sampler updates the sample through iteratively denoising to the original sample, *i.e.*,

$$\hat{\mathbf{x}}_0 = \mathbf{x}_t - t\mathbf{u}_\theta(\mathbf{x}_t, t, 0)$$
$$\mathbf{x}_r = (1 - r)\hat{\mathbf{x}}_0 + r\boldsymbol{\epsilon},$$

where $\boldsymbol{\epsilon} \sim \mathcal{N}(\mathbf{0}, \mathbf{I})$ is a random Gaussian noise. To facilitate the best of both Euler sampler and restart sampler, one can use the stochastic sampler proposed in CTM (Kim et al., 2024). At high-level, CTM sampler denoises to the intermediate timestep $s \in [0, r]$, then shift the input through forward processes. Formally, this can be written as

$$s \leftarrow g(r, \gamma)$$
$$\mathbf{x}_s \leftarrow \mathbf{x}_t + (s - t)\mathbf{u}_\theta(\mathbf{x}_t, t, s)$$
$$\mathbf{x}_r \leftarrow \frac{\alpha_r}{\alpha_s}\mathbf{x}_s + \left(\sigma_r - \sigma_s\frac{\alpha_r}{\alpha_s}\right)\boldsymbol{\epsilon},$$

where $g : [0, 1] \times \mathbb{R} \to [0, 1]$ is a time shifting function that shifts $r$ to intermediate timestep $s$ with stochasticity $\gamma$, and $\boldsymbol{\epsilon} \sim \mathcal{N}(\mathbf{0}, \mathbf{I})$ is a random Gaussian. Note that if $s = 0$, it becomes restart sampler, and if $s = r$, it becomes Euler sampler. Following CTM (Kim et al., 2024), we consider the function $s$ that satisfies

$$\frac{s}{1 - s} = (1 - \gamma)\frac{r}{1 - r} \quad \Rightarrow \quad s = \frac{(1 - \gamma) \cdot r}{1 - \gamma \cdot r},$$

where $\gamma \in [0, 1]$ is a stochasticity hyperparameter. Then we have $\frac{\alpha_r}{\alpha_s} = 1 - \gamma \cdot r$ and $\sigma_r - \sigma_s\frac{\alpha_r}{\alpha_s} = \gamma \cdot r$. One can see that if $\gamma = 1$, we have $s = 0$ (restart sampler), and if $\gamma = 0$, we have $s = r$ (Euler sampler). We find that varying $\gamma$ improves $\mathrm{FD}_{\mathrm{DINOv2}}$ and IS, while increasing FID.

**Time-distribution shift.** Note that for denoising with higher resolution, it has been shown that shifting the timestep distribution during sampling enhances the fidelity (Esser et al., 2024). Specifically, let $s$ be shifting factor, then the shifted time is given by

$$t_{\mathrm{shift}} = \frac{st}{1 + (s - 1)t}, \tag{14}$$

for $t > 0$. In practice, we use uniform time discretization as default and use shift $s = 1.0$ (*i.e.*, no shift) for ImageNet 256×256 models, and shift $s = 1.5$ for ImageNet 512×512.

# D    EVALUATION

We follow the setup in EDM2 (Karras et al., 2024b) for evaluation. For evaluation, we use NVIDIA H200 GPUs to generate 50,000 samples to compute FID, Inception score, and $FD_{DINOv2}$. Note that we use FP32 precision for generation, while using BF16 shows negligible differences. Here, we briefly overview each metric we used for the evaluation:

- **FID** (Heusel et al., 2017) evaluates the fidelity of generated samples through comparing the feature distances using the Inception-v3 (Szegedy et al., 2016) model. Specifically, we first gather the embeddings of 1.3M training images as well as generated images. Then we compute the feature distance by fitting into the multivariate Gaussian distribution.
- **Inception Score (IS)** (Salimans et al., 2016) measure the image quality and diversity using the Inception-v3 classifier. Specifically, we compute the cross-entropy between the ground-truth label and the logit output of classifier to compute IS.
- **Fréchet Distance with DINOv2 ($FD_{DINOv2}$)** (Stein et al., 2023) also measures the fidelity of generated images, but using DINOv2 (Oquab et al., 2023) as feature encoder. $FD_{DINOv2}$ is shown to be more aligned with human perception. Note that the computation is sames as FID, but differs by changing Inception-v3 to DINOv2-L/14.

# E    DETAILED QUANTITATIVE RESULTS

We provide additional evaluation results for flow models (SiT-XL/2+REPA) and flow maps (DMF-XL/2+). First, we show the reproduced results for flow models, and show additional ablation studies on the DMF decoder depth, MG scale, time proposal distribution, and effect of QK normalization for DMF-XL/2+ modeels.

**Evaluation of flow models.** We show the evaluation results of the flow models (SiT-XL/2+REPA). For $256 \times 256$ resolution, we re-use the pretrained model from their official repository[3], and for $512 \times 512$ resolution, we trained the model for 400 epochs (see Tab. 6). In Tab. 7, we report the FID and $FD_{DINOv2}$ of each $256 \times 256$ and $512 \times 512$ resolution model by varying the choice of samplers (SDE and ODE), CFG scale, and number of denoising steps (as well as total NFE). Note that REPA used Euler-Maruyama SDE sampler with 250 steps, CFG scale 1.8 and guidance interval (Kynkäänniemi et al., 2024) on $[t_{min}, t_{max}] = [0.0, 0.7]$ as default, which achieves FID=1.42. On the other hand, our reproduction achieves FID=1.41 when using CFG scale of 2.0, and achieves better FID of 1.37 when using ODE sampler (Heun's method) with 128 steps (total NFE=434).

Similarly, for ImageNet $512 \times 512$, the original paper reported FID=2.08 when trained for 200 epochs by using CFG=1.35, SDE sampler with 250 steps. For our reproduction, we achieve FID=1.85 when using same configurations, while achieves better FID=1.37 by using Heun's method with CFG=2.0, and guidance interval technique.

Table 7: FID and $FD_{DINOv2}$ results of SiT-XL/2+REPA models on ImageNet $256 \times 256$ and $512 \times 512$. The gray colored rows are excerpted from original REPA paper.

| Epochs | Resolution | Sampler | CFG | $[t_{min}, t_{max}]$ | # Steps | NFE | FID↓ | $FD_{DINOv2}$↓ |
|--------|-----------|---------|-----|----------------------|---------|-----|------|----------------|
| 800 | $256 \times 256$ | SDE | 1.8 | [0.0, 0.7] | 250 | 425 | 1.42 | - |
| 800 | $256 \times 256$ | SDE | 1.8 | [0.0, 0.7] | 250 | 425 | 1.53 | 71.6 |
| 800 | $256 \times 256$ | SDE | 2.0 | [0.0, 0.7] | 250 | 425 | 1.41 | **62.4** |
| 800 | $256 \times 256$ | ODE | 2.0 | [0.0, 0.7] | 128 | 434 | **1.37** | 62.6 |
| 200 | $512 \times 512$ | SDE | 1.35 | [0.0, 1.0] | 250 | 500 | 2.08 | - |
| 400 | $512 \times 512$ | SDE | 1.35 | [0.0, 1.0] | 250 | 500 | 1.85 | 41.5 |
| 400 | $512 \times 512$ | SDE | 1.8 | [0.0, 0.8] | 250 | 449 | 1.45 | 36.6 |
| 400 | $512 \times 512$ | ODE | 2.0 | [0.0, 0.8] | 128 | 460 | **1.37** | **32.0** |

---

[3] https://github.com/sihyun-yu/REPA

**Evaluation of DMF models.** Next, we provide additional ablation on the effect of depth, MG scale, time proposal distribution, and effect of QK-normalization for DMF-XL/2+ models. Tab. 8 shows the results. First, we observe that using MG scale of 0.6 generally shows better performance than MG scale of 0.5. Note that the flow models achieve the best performance when CFG scale of 2.0, which corresponds to MG scale of 0.5. On the other hand, DMF model favors slightly higher guidance scale, which we suspect that the few-step models require higher guidance scale to generate high-fidelity sample within few-steps. Furthermore, we observe that using depth of 20 is better than 22. Note that the base flow models converted to DMF model without fine-tuning achieved the best performance when using depth of 22. However, we observe that using more layers for decoder generally improves the performance. Lastly, we observe that using more aggressive time proposal distribution improves 1-step performance, which is consistent with our observation in Sec. A.2.

For ImageNet $512 \times 512$ model, we observe that QK normalization significantly helps stabilizing the training, and improves the overall performance. Note that we did not find any gain when applying QK normalization to 256 resolution models. Furthermore, we find that it suffices to train for 400K iterations for DMF-XL/2+ 256 resolution model (no performance gain for longer training), while 512 resolution model keep improves its performance for longer training iterations. We find that 700K training iterations show good convergence.

Table 8: FID and $\mathrm{FD_{DINOv2}}$ evaluation of DMF-XL/2+ models on ImageNet $256 \times 256$. We vary dmf depth $d$, MG scale $\omega$, and time proposal $(\mu_{\mathrm{MF}}^{(1)}, \mu_{\mathrm{MF}}^{(2)})$. We report 1, 2, and 4-step (NFE=1,2,4) FID and $\mathrm{FD_{DINOv2}}$ for each training run.

| Iter. | $d$ | $\omega$ | $(\mu_{\mathrm{MF}}^{(1)}, \mu_{\mathrm{MF}}^{(2)})$ | QK-norm | 1-step (NFE = 1) | | 2-step (NFE = 2) | | 4-step (NFE = 4) | |
|---|---|---|---|---|---|---|---|---|---|---|
| | | | | | FID | $\mathrm{FD_{DINOv2}}$ | FID | $\mathrm{FD_{DINOv2}}$ | FID | $\mathrm{FD_{DINOv2}}$ |
| *DMF-XL/2+ 256×256* | | | | | | | | | | |
| 200K | 22 | 0.5 | (-0.4, -0.4) | ✗ | 2.91 | 162.9 | 1.82 | 92.8 | 1.70 | 78.0 |
| 200K | 22 | 0.55 | (-0.4, -0.4) | ✗ | 2.74 | 154.3 | 1.74 | 83.1 | 1.55 | 68.7 |
| 200K | 22 | 0.6 | (-0.4, -0.4) | ✗ | 2.60 | 143.9 | 1.73 | 73.0 | 1.54 | 60.1 |
| 200K | 20 | 0.55 | (-0.4, -0.4) | ✗ | 2.54 | 145.8 | 1.67 | 82.5 | **1.50** | 69.9 |
| 200K | 20 | 0.6 | (-0.4, -0.4) | ✗ | 2.46 | 133.0 | 1.71 | 71.5 | 1.50 | 60.2 |
| 200K | 20 | 0.55 | (0.4, -1.2) | ✗ | 2.50 | 140.4 | 1.68 | 85.7 | 1.60 | 72.6 |
| 200K | 20 | 0.6 | (0.4, -1.2) | ✗ | 2.25 | 128.2 | 1.69 | 74.6 | 1.53 | 63.7 |
| 400K | 20 | 0.6 | (0.4, -1.2) | ✗ | **2.16** | **122.3** | **1.64** | **69.8** | 1.51 | **59.9** |
| *DMF-XL/2+ 512×512* | | | | | | | | | | |
| 200K | 20 | 0.6 | (0.4, -1.2) | ✗ | 2.97 | 93.5 | 2.01 | 54.8 | 1.87 | 43.2 |
| 400K | 20 | 0.6 | (0.4, -1.2) | ✗ | 2.71 | 86.9 | 1.95 | 52.3 | 1.80 | 41.6 |
| 200K | 20 | 0.6 | (0.4, -1.2) | ✔ | 2.50 | 84.2 | 1.94 | 53.3 | 1.84 | 44.2 |
| 400K | 20 | 0.6 | (0.4, -1.2) | ✔ | 2.28 | 77.7 | 1.87 | 50.9 | 1.79 | 42.1 |
| 700K | 20 | 0.6 | (0.4, -1.2) | ✔ | **2.12** | **72.3** | **1.75** | **43.9** | **1.68** | **39.5** |

**CTM-$\gamma$ sampler.** We provide additional evaluation of DMF-XL/2+ model when using stochastic CTM sampler. We follow the setup illustrated in Appendix C. As shown in Tab. 9, we observe that Euler sampler (*i.e.*, $\gamma$=0) achieves the lowest FID, while adding stochasticity improves $\mathrm{FD_{DINOv2}}$ and IS, *e.g.*, $\gamma$=0.96 achieves lowest $\mathrm{FD_{DINOv2}}$=47.9, and $\gamma$=0.98 achieves highest IS=328.0. Thus, adding stochasticity through CTM sampler helps generating diverse samples, and improves semantic fidelity, while deterministic Euler sampler achieves the best FID, which is consistent to Fig. 5.

Table 9: NFE-4 CTM sampler with various $\gamma$.

| $\gamma$ | FID↓ | $\mathrm{FD_{DINOv2}}$ ↓ | IS↑ |
|---|---|---|---|
| 0.00 | **1.51** | 60.0 | 280.1 |
| 0.01 | 1.75 | 55.8 | 283.2 |
| 0.02 | 2.12 | 53.8 | 283.1 |
| 0.04 | 3.12 | 55.7 | 274.0 |
| 0.10 | 7.79 | 93.8 | 219.3 |
| 0.50 | 31.7 | 335.5 | 83.3 |
| 0.90 | 5.55 | 61.6 | 291.6 |
| 0.96 | 3.03 | **47.9** | 326.8 |
| 0.98 | 2.39 | 50.0 | **328.0** |
| 0.99 | 2.17 | 52.8 | 326.6 |
| 1.00 | 1.99 | 56.8 | 322.7 |

## F    ADDITIONAL OBSERVATION

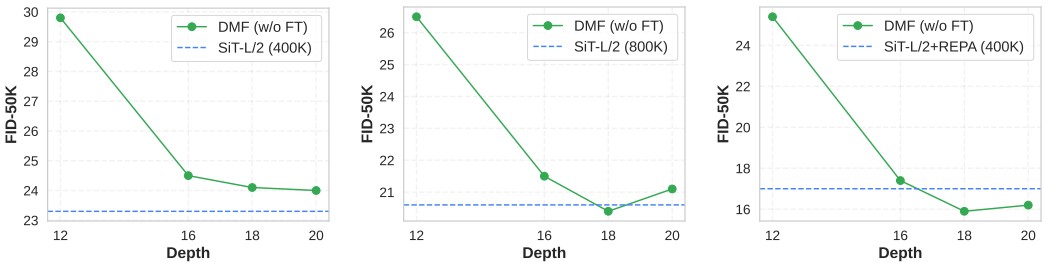

Figure 8: **Strong flow models are better flow map.** We compare the gap between flow model and DMF converted flow map by varying the flow model. We show results for SiT-L/2 trained for 400K and 800K iterations, and SiT-L/2+REPA trained for 400K iterations. FID-50K is reported by sampling with 16-step Euler sampler without using CFG.

As shown in Fig. 3, the DMF model converted from pretrained flow model often outperforms flow model without fine-tuning. We further analyze this phenomenon in details. To study, we exploit SiT-L/2 models trained for 400K and 800K iterations, and SiT-L/2+REPA model trained for 400K iterations. We vary the depth of DMF model by 12, 16, 18, and 20, similar to the setup in Tab. 1.

As shown in Fig. 8, we find that the DMF models from SiT-L/2 400K model do not show gain in compared to their base model. However, as training proceeds, the DMF model from SiT-L/2 800K starts to outperform its base model by setting appropriate depth. Furthermore, we notice that DMF model significantly outperforms when initialized from SiT-L/2+REPA 400K model, even trained for only 400K iterations. We hypothesize that this is due to the representational capacity of base flow model in turning to flow map. As training proceeds, it is well-known that the model forms a representational knowledge, thus SiT-L/2 800K model tends to be better flow map than SiT-L/2 400K model. On the other hand, if we explicitly align the model with self-supervised representation, the model quickly forms the representation, which helps to form better flow map.

## G    DISCUSSION

**Limitations.**   Although our method demonstrates substantial improvements in FID, IS, and $FD_{DINOv2}$, we frequently observe visual artifacts in generated samples, particularly under the 1-step setting. We attribute this issue partly to the constraints of our current experimental setup, which relies on the VAE latent space and the ImageNet dataset, both of which contain inherent quality limitations. As a next step, it would be valuable to validate the effectiveness of DMF on large-scale text-to-image (Esser et al., 2021) and text-to-video (Wan et al., 2025) models.

**Future directions.**   We believe our approach opens a promising line of research toward efficient training and inference of diffusion and flow models. For example, reducing inference cost may enable a re-examination of scaling laws for diffusion models (Esser et al., 2024; Blattmann et al., 2023; Yin et al., 2025), allowing more compute to be allocated per denoising step. Another promising direction is inference-time scaling (Ma et al., 2025), such as searching over initial or intermediate noise states using Restart solvers. Finally, extending post-training algorithms, which have so far been mainly studied for diffusion and flow models (Wallace et al., 2024; Lee et al., 2025; Liu et al., 2025; Xue et al., 2025), to flow maps remains an open challenge.

# H QUALITATIVE EXAMPLES

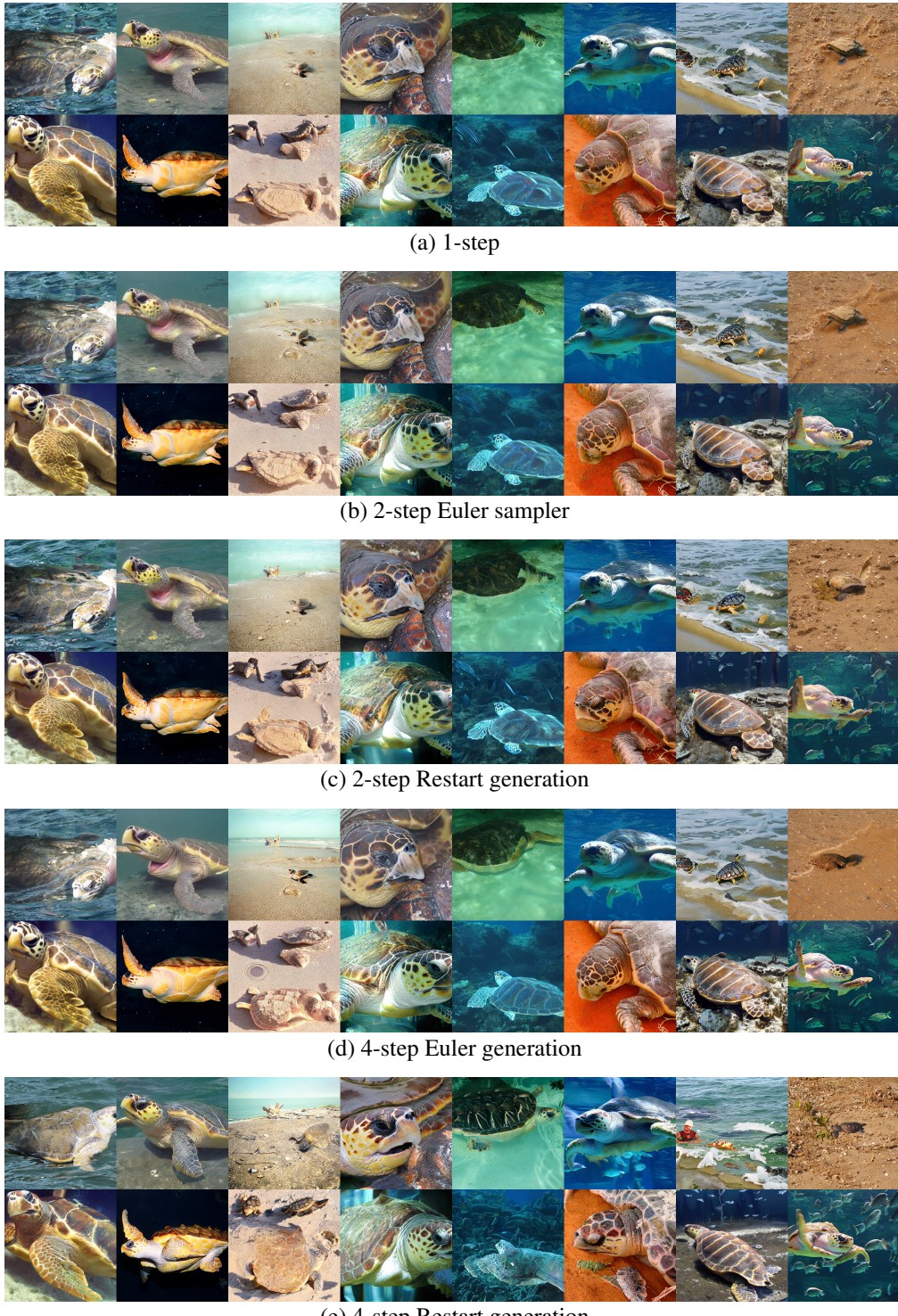

(a) 1-step

(b) 2-step Euler sampler

(c) 2-step Restart generation

(d) 4-step Euler generation

(e) 4-step Restart generation

Figure 9: Generation with DMF-XL/2+-512 with class id 33: `loggerhead turtle`

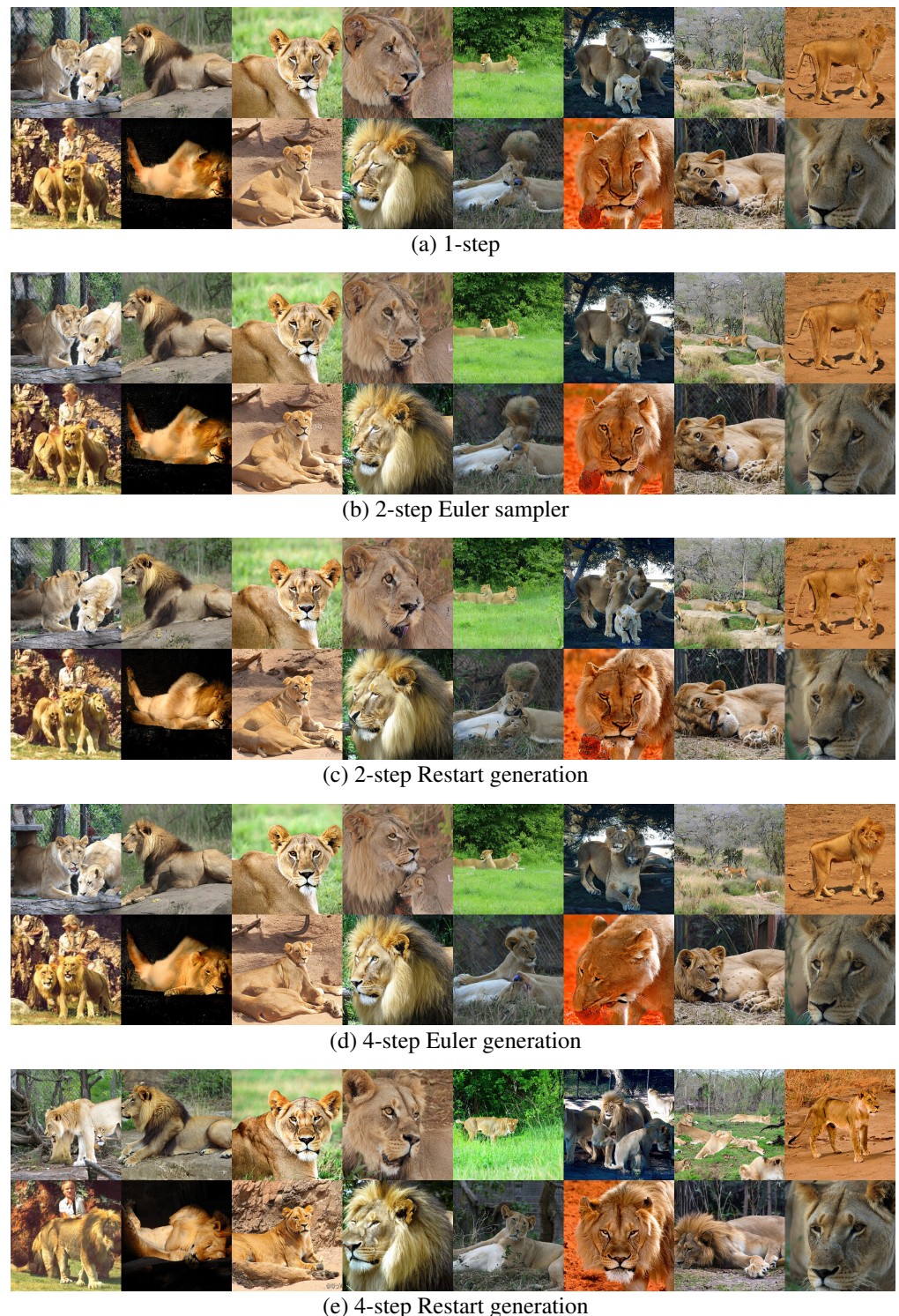

(a) 1-step

(b) 2-step Euler sampler

(c) 2-step Restart generation

(d) 4-step Euler generation

(e) 4-step Restart generation

Figure 10: Generation with DMF-XL/2+-512 with class id 291: `lion`

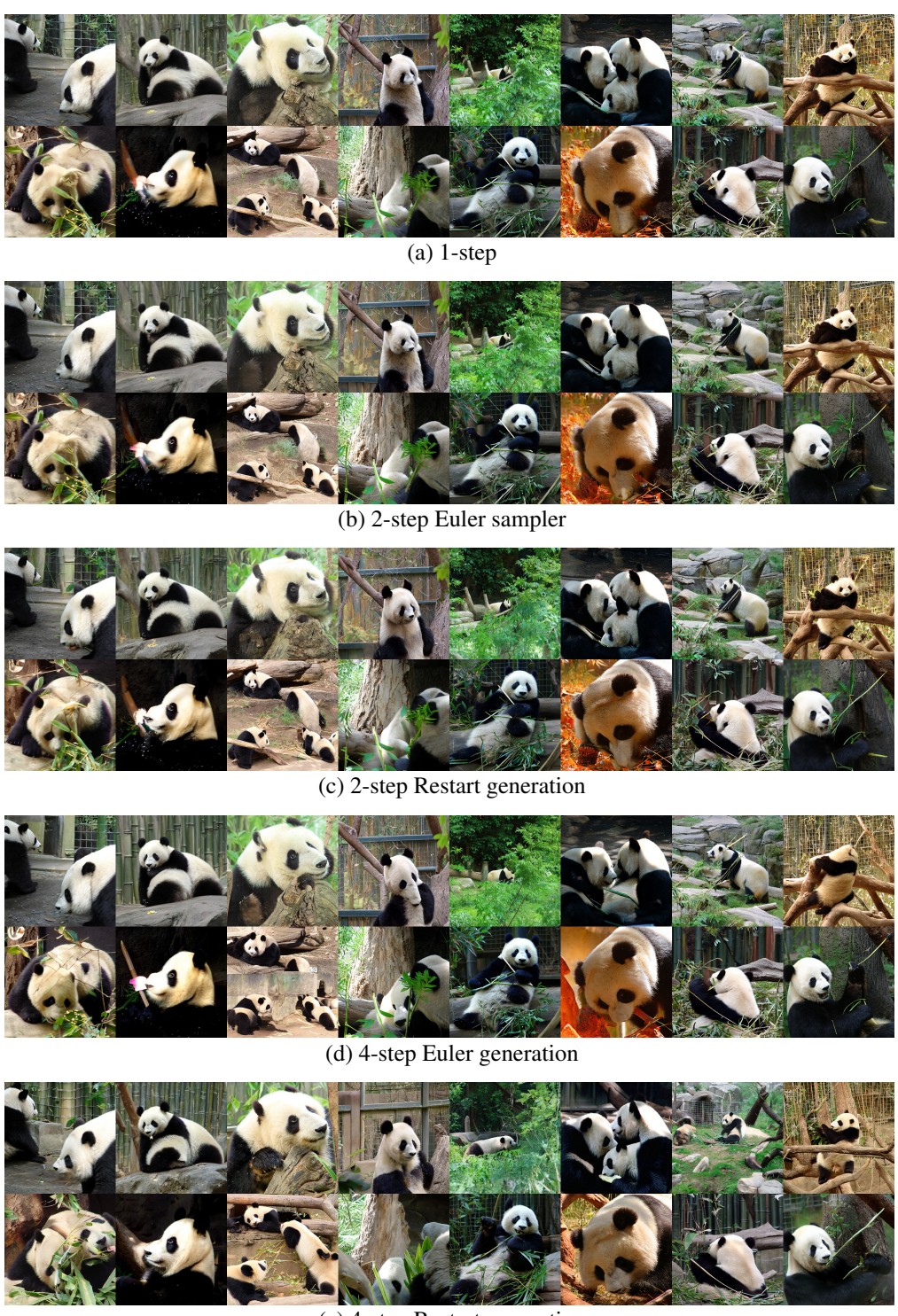

(a) 1-step

(b) 2-step Euler sampler

(c) 2-step Restart generation

(d) 4-step Euler generation

(e) 4-step Restart generation

Figure 11: Generation with DMF-XL/2+-512 with class id 388: `panda`

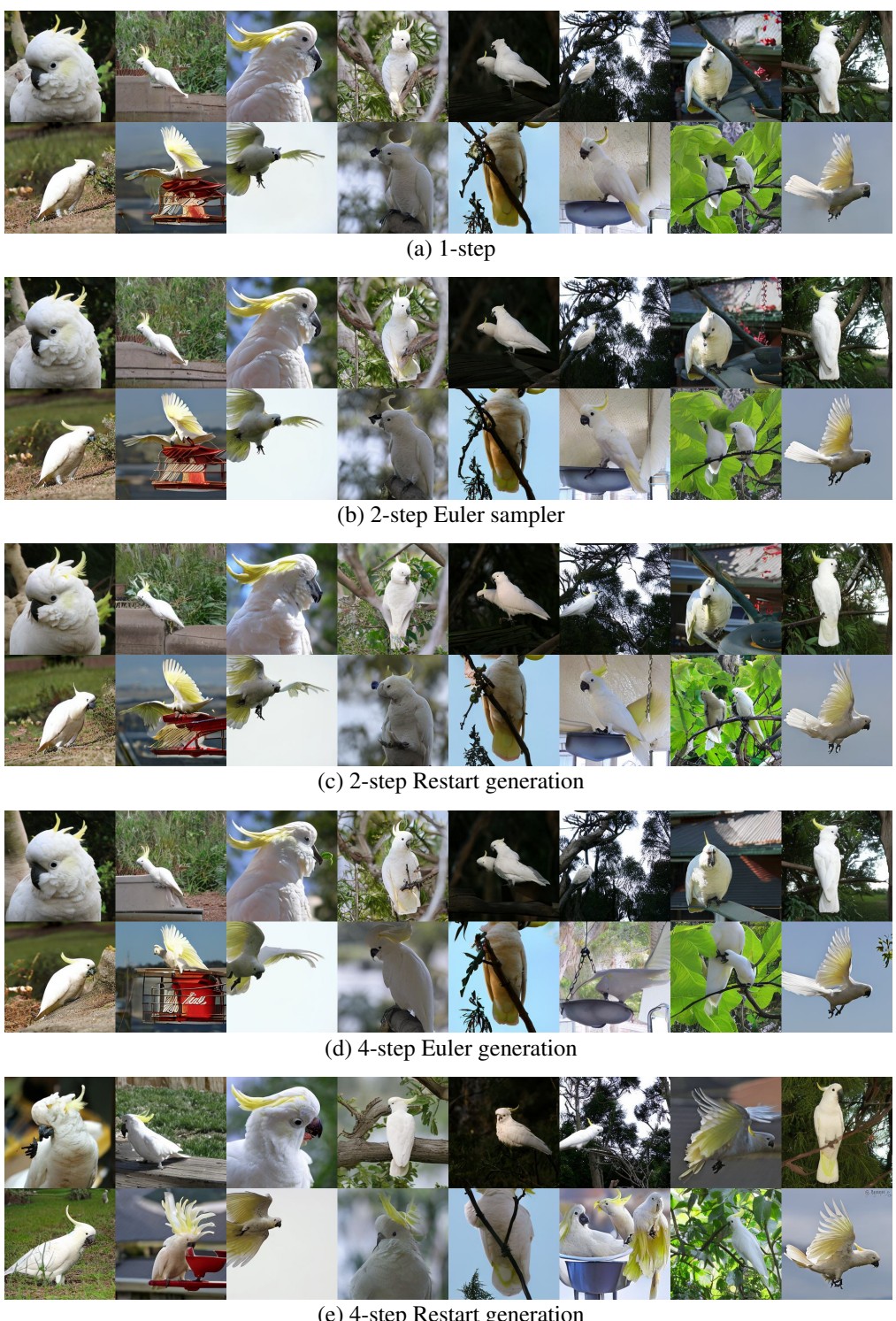

(a) 1-step

(b) 2-step Euler sampler

(c) 2-step Restart generation

(d) 4-step Euler generation

(e) 4-step Restart generation

Figure 12: Generation with DMF-XL/2+-256 with class id 89: `sulphur-crested cockatoo`

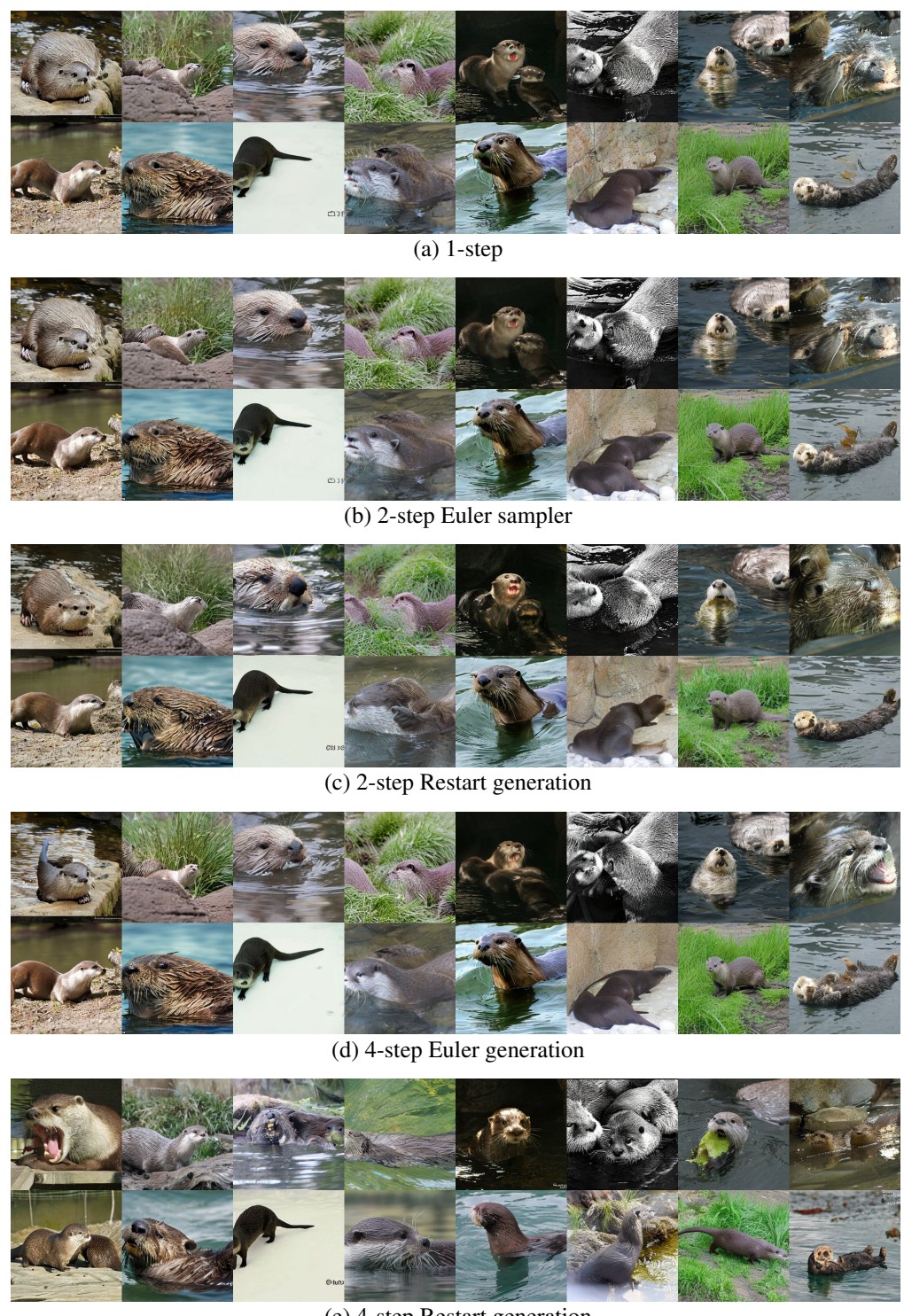

(a) 1-step

(b) 2-step Euler sampler

(c) 2-step Restart generation

(d) 4-step Euler generation

(e) 4-step Restart generation

Figure 13: Generation with DMF-XL/2+-256 with class id 360: `otter`

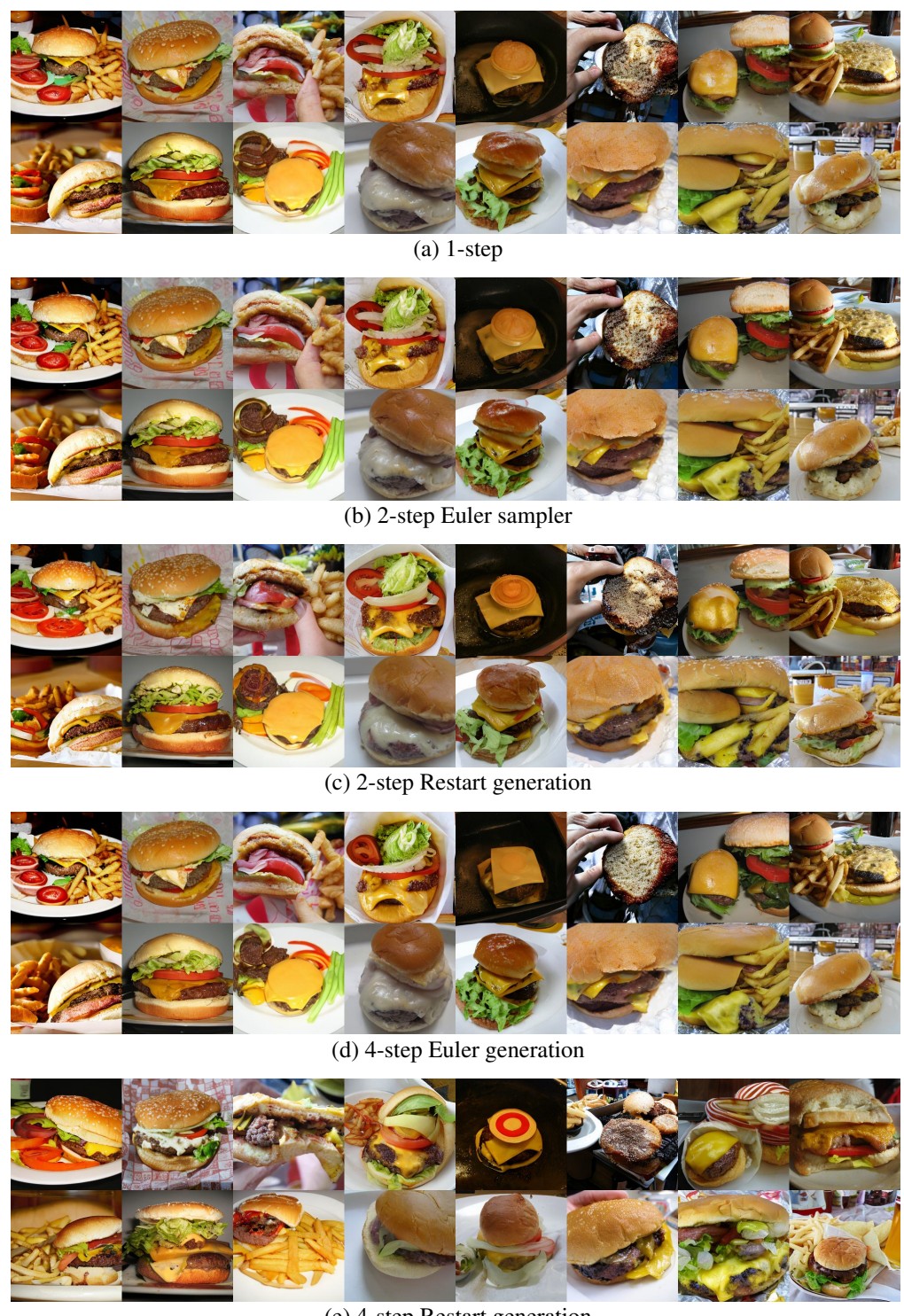

(a) 1-step

(b) 2-step Euler sampler

(c) 2-step Restart generation

(d) 4-step Euler generation

(e) 4-step Restart generation

Figure 14: Generation with DMF-XL/2+-256 with class id 933: `cheeseburger`

