# OpenReview forum: "Decoupled MeanFlow: Turning Flow Models into Flow Maps for Accelerated Sampling"
_ICLR.cc/2026/Conference — ICLR 2026 Poster_

### Official Review · Reviewer_9VuX · 2025-10-29

**Soundness:** 4
**Presentation:** 3
**Contribution:** 3
**Rating:** 6
**Confidence:** 4

**Summary:**

The paper proposes Decoupled MeanFlow (DMF), a way to convert a pretrained flow (or diffusion-as-flow) model into a flow-map model without changing the backbone: treat the early blocks as an encoder conditioned on the current time $t$ and the late blocks as a decoder conditioned on the next time $r$. This decoupling lets the model learn the average velocity between $(t,r)$ while reusing the original timestep embedding and velocity head. With a flow-matching warm-up then flow-map fine-tuning (plus model-guidance and a robust Cauchy loss), DMF targets ultra-few-step sampling (1–4 steps).

**Strengths:**

1. This paper proposes simple, compatible architecture. The key idea—drop $r$ from the encoder and $t$ from the decoder, i.e., $u_\theta(x_t,t,r)=g_\theta(f_\theta(x_t,t),r)$—is elegant and backbone-agnostic, enabling plug-and-play conversion of existing flow models.


2. Strong empirical case that “your flow model is secretly a flow map.” Without any finetuning, the converted model can already outperform the base SiT in FID at equal step counts; with decoder-only finetuning it gets further gains. The depth ablation is informative.


3. State-of-the-art 1–4 step results under fair comparisons. Tables show consistent wins over MeanFlow baselines at comparable training budgets, and competitive/best few-step FIDs on ImageNet 256/512.

**Weaknesses:**

See Questions

**Questions:**

1. Table 1 suggests both model guidance (MG) and REPA are crucial for DMF’s final performance. DMF typically splits at later layers (e.g., layer 18), whereas REPA often targets earlier layers (e.g., layer 8 in SiT-L/2). Is there a principled relationship between the REPA alignment layer and the DMF encoder–decoder split?
2. The paper mentions visual artifacts in generated outputs in limitations. Could you include representative ‘bad-case’ examples to illustrate these failures?

---

> ### Author Response · Authors · 2025-11-21
> **Response to Reviewer 9VuX**
>
> Dear reviewer 9VuX,
>
> We sincerely appreciate your valuable comments. We address each of your questions and concerns one-by-one in what follows.
>
> ---
> ### Q1. Relationship between REPA alignment layer and the DMF encoder–decoder split layer
> We thank the reviewer for raising this intriguing question. In short, REPA and DMF target complementary aspects of representation quality, but they are not required to be aligned to the same layer. Specifically, REPA improves early-to-middle encoder representations by enforcing alignment between latent features at a specific internal block (e.g., 8th layer). In contrast, DMF is concerned with the functional division of roles between encoder and decoder, e.g., early blocks focus on representing $x_t$, while late blocks predict the transition toward $r$. These roles naturally emerge around deeper layers (e.g., 16–20), consistent with prior analyses in Figure 3.
>
> Empirically, we observe that:
> - REPA improves both MF and DMF (while DMF shows larger gain; Table 1)
> - DMF conversion without finetuning also benefits from REPA (Appendix C).
>
> This suggests a principled relationship that REPA enhances the quality of the encoder representation, and DMF is the architecture best positioned to exploit this improved representation because it isolates the encoder’s role. The alignment layer used in REPA does not need to match the DMF split depth as REPA strengthens the whole representational hierarchy, and DMF leverages it by allocating the encoder and decoder to distinct tasks. We will clarify this point in the final version.
>
> ---
> ### W2. Visual artifacts.
>
> While we have achieved a reasonable FID score, we observe that there are some artifacts for 1-step generation results. We depicted qualitative results in Appendix, where 1-step results are quite inferior to 2-step or 4-step results. We suspect that this is an inherent limitation of current dataset and VAEs, where such artifacts could be removed by 1) scaling up the model size and dataset, and 2) using a better tokenizer that better represents the image.

---

> > ### Author Response · Authors · 2025-11-28
> >
> > Dear Reviewer 9VuX,
> >
> > As the discussion period concludes in the coming week, we would like to offer a gentle reminder in case you have any remaining questions or concerns. We hope that our rebuttal has addressed your points, but please feel free to let us know if any clarification would be helpful.
> >
> > Thank you for your time and consideration.
> >
> > Sincerely,
> > The Authors

---

### Official Review · Reviewer_1Fiu · 2025-10-29

**Soundness:** 3
**Presentation:** 3
**Contribution:** 2
**Rating:** 6
**Confidence:** 5

**Summary:**

The authors explore architectural improvements for training flow map models and propose a series of techniques to enhance the performance of MeanFlow under comparable compute budgets. The resulting Decoupled MeanFlow (DMF) framework introduces a decoupled encoder–decoder design that enables pretrained flow matching models to be seamlessly converted into flow map models without architectural modification. By conditioning the decoder on the next timestep while keeping the encoder focused on the current one, DMF improves both efficiency and transferability of pretrained representations. This approach, combined with flow-matching warm-up, adaptive Cauchy loss, and tailored timestep sampling, achieves state-of-the-art few-step generative performance on ImageNet 256×256, reaching a 1-step FID of 2.16 and matching standard flow model performance at 4 steps.

**Strengths:**

1. **Training-free flow map transformation**

The proposed decoupled architecture allows pretrained flow models to be directly repurposed as flow maps without additional fine-tuning, which is both conceptually elegant and practically impactful. This demonstrates a viable paradigm for transformer-based flow map models that leverages existing large-scale flow model checkpoints, reducing training cost and broadening applicability.

2. **Broader applicability of the fine-tuning paradigm**

The proposed fine-tuning strategy has strong practical implications. Because DMF can be directly initialized from large-scale pretrained flow checkpoints (e.g., FLUX-dev, SD3, etc.), it enables transforming existing models into few-step generators with a modest additional compute budget. This makes the method immediately applicable in real-world or large-foundation-model scenarios where retraining from scratch is infeasible.

3. **Comprehensive empirical validation**

The paper includes detailed ablations, consistently validating the design claims, particularly the benefit of decoupling timestep conditioning and the effectiveness of flow-matching warm-up for stability and efficiency.

**Weaknesses:**

1. **Stability of the JVP term**

The proposed method does not directly address the well-known stability issue of the JVP term. This instability has been repeatedly identified as the primary bottleneck in scaling consistency-based methods to large-scale applications such as text-to-image or text-to-video generation (Lu & Song, 2024; Chen et al., 2025; Zheng et al., 2025). Therefore, while the techniques presented in the paper for improving MeanFlow training remain valuable, the overall scope of the paper remains somewhat limited. It would nonetheless strengthen the paper to include an empirical comparison of training stability between DMF and vanilla MeanFlow (e.g., visualizations of training losses, gradient norms, or JVP magnitudes).

2. **Insufficient ablations**

The use of the adaptive weighted Cauchy loss (Eq. 6) for MeanFlow optimization is intriguing but under-justified. The paper should better explain why this choice improves robustness beyond prior alternatives (e.g., the adaptive normalization proposed in Geng et al., 2024 and the tangent normalization proposed in Lu & Song, 2024) with supporting evidence. Also, the paper should cite Dao et al., 2025, as they also proposed applying a near identical Cauchy loss to consistency training.

3. **Limited architectural exploration**

While the encoder-decoder design for training-free flow map transformation is intriguing, the current experiments primarily rely on the vanilla DiT. It remains unclear whether the encoder–decoder decoupling generalizes to other variants of DiT (i.e. LightningDiT (Yao et al., 2025) or DDT (Wang et al., 2025)), and the paper would be greatly strengthened from some discussions of this.

Lu, Cheng, and Yang Song. "Simplifying, stabilizing and scaling continuous-time consistency models." arXiv preprint arXiv:2410.11081 (2024).

Chen, Junsong, Shuchen Xue, Yuyang Zhao, Jincheng Yu, Sayak Paul, Junyu Chen, Han Cai, Song Han, and Enze Xie. "Sana-sprint: One-step diffusion with continuous-time consistency distillation." arXiv preprint arXiv:2503.09641 (2025).

Zheng, Kaiwen, Yuji Wang, Qianli Ma, Huayu Chen, Jintao Zhang, Yogesh Balaji, Jianfei Chen, Ming-Yu Liu, Jun Zhu, and Qinsheng Zhang. "Large Scale Diffusion Distillation via Score-Regularized Continuous-Time Consistency." arXiv preprint arXiv:2510.08431 (2025).

Geng, Zhengyang, Ashwini Pokle, William Luo, Justin Lin, and J. Zico Kolter. "Consistency models made easy." arXiv preprint arXiv:2406.14548 (2024).

Wang, Shuai, Zhi Tian, Weilin Huang, and Limin Wang. "Ddt: Decoupled diffusion transformer." arXiv preprint arXiv:2504.05741 (2025).

Yao, Jingfeng, Bin Yang, and Xinggang Wang. "Reconstruction vs. generation: Taming optimization dilemma in latent diffusion models." In Proceedings of the Computer Vision and Pattern Recognition Conference, pp. 15703-15712. 2025.

Dao, Quan, Khanh Doan, Di Liu, Trung Le, and Dimitris Metaxas. "Improved training technique for latent consistency models." arXiv preprint arXiv:2502.01441 (2025).

**Questions:**

1. **Representation quality**

Does the proposed decoupled architecture improve the representation quality of the transformer encoder itself compared to vanilla MeanFlow (e.g., linear probing accuracy) under similar training budget?

2. **Diminishing gains with REPA**

The paper notes that REPA fine-tuning yields diminishing returns (Tab. 1). Could the authors provide quantitative comparisons (apply REPA only for MF fine-tuning or apply REPA at both stages) or intuition on why representation alignment offers less benefit under the DMF design or consistency training in general?

3. **Loss formulation**

When doing MF fine-tuning, the authors use sum of MF & FM losses instead of batch splitting proposed in MeanFlow. However, the adaptive weighting function itself breaks the "correctness" of the FM objective (e.g., the optimal solution is no longer $\mathbb{E}[v | x_t = x]$). Have the authors evaluated DMF performance with the vanilla flow-matching loss without such weighting for ablation, and how does it compare in stability and convergence?

4. **Timestep sampling**

Have the authors experimented with alternative timestep proposal paradigms (e.g., $p(t, r) = p(t)p(r | t)$)?

---

> ### Author Response · Authors · 2025-11-21
> **Response to Reviewer 1Fiu**
>
> Dear reviewer 1Fiu,
>
> We sincerely appreciate your valuable comments. We address each of your questions and concerns one-by-one in what follows.
>
> ---
> ### W1. Stability of the JVP term
> We would like to clarify that our approach is not designed to resolve the instability issue of JVP computation. While JVP stability is an important issue in training flow maps, we found that the stability issues have to be more related to the architectural design rather than our decoupling scheme. From our observation, we didn’t observe any instability issues for both MF and DMF models when training on ImageNet-256 dataset. On the other hand, for ImageNet-512, we observed some abnormal gradient norms during training, which happens in both Flow Matching (FM) loss and MeanFlow (MF) loss.
>
> Specifically, we extended the SiT-XL/2+REPA ImageNet-512 training from 240 epochs to 400 epochs, which achieves FID=1.37. Then, we fine-tune the DMF model with the same setup as before. However, we observe that there are some instability issues during training, which shows inferior performance than the DMF model fine-tuned from SiT-XL/2+REPA 240 epoch checkpoint. Thus, similar to SANA-sprint [1], we find that adding QK normalization [2] to the Attention layers alleviate such issues, and stabilize the training that obtains better results (see Table below). In addition, if we train longer (700K), we find that our model achieves 1-step FID=2.12 and 2-step FID=1.75, which surpasses the previous state-of-the-art sCD-XXL [3], which has 1-step FID=2.28 and 2-step FID=1.88.
>
> \begin{array}{lllcccc}
> \hline
> \text{FM PT Ep.} & \text{DMF FT Iter} & \text{QK-norm} & \text{1-step FID} & \text{1-step FDD}& \text{2-step FID} & \text{2-step FDD}\newline
> \hline
> \text{240 } & \text{200K} & \text{False}  & \text{2.58} & \text{90.8} & \text{2.13} & \text{60.6} \newline
> \text{240 } & \text{400K} & \text{False}  & \text{2.36} & \text{82.5} & \text{2.04} & \text{57.6} \newline
> \hline
> \text{400 } & \text{200K} & \text{False}  & \text{2.97} & \text{93.5} & \text{2.01} & \text{54.8} \newline
> \text{400 } & \text{400K} & \text{False}  & \text{2.71} & \text{86.9} & \text{1.95} & \text{52.3} \newline
> \hline
> \text{400 } & \text{200K} & \text{True}   & \text{2.50} & \text{84.2} & \text{1.94} & \text{53.3} \newline
> \text{400 } & \text{400K} & \text{True}   & \text{2.28} & \text{77.7} & \text{1.87} & \text{50.9} \newline
> \text{400 } & \text{700K} & \text{True}   & \text{2.12} & \text{72.3} & \text{1.75} & \text{43.9} \newline
> \hline
> \end{array}
>
> Besides practical issues, note that the DMF model can achieve a lower JVP variance due to the decoupling scheme. Specifically, let us denote $u(x_t,t,r) = g(f(x_t,t), r)$, where $f$ is an encoder and $g$ is a decoder. Then the JVP term can be written as
> $$
> \frac{du}{dt} = \frac{dg}{d h_t}\frac{df}{dh_t} = \frac{dg}{dh_t} (\frac{dx_t}{dt} \frac{\partial f}{\partial x_t} + \frac{\partial f}{\partial t}).
> $$
> On the other hand, for MF model, we have $u(x_t,t,r) = g(f(x_t,t,r),t,r)$, thus we have
> $$
> \frac{du}{dt} = \frac{\partial g}{\partial h_t}\frac{df}{dh_t} + \frac{\partial g}{\partial t} = \frac{dg}{dh_t} (\frac{dx_t}{dt} \frac{\partial f}{\partial x_t} + \frac{\partial f}{\partial t}) + \frac{\partial g}{\partial t}.
> $$
> Therefore, MF model requires additional JVP computation on the decoder, thus might be more fragile if we use a bigger decoder or different diffusion transformer architecture. For instance, which we will describe in W3 below, we find the fine-tuning Decoupled Diffusion Transformer (DDT) [4] model with MF is very unstable which we cannot resolve the instability issue without using QK normalization, while DMF can be trained without instability issue. As such, we believe our approach can be effectively applied to the large-scale image (e.g., text-to-image) or video (e.g., text-to-video or image-to-video) models, which we leave as future work.
>
> ---
> ### W2. Ablation on Cauchy loss.
>
> Thanks for suggesting the relevant works. Similar to the findings in sCM [3], we find that using adaptive weighting helps the performance for 1- and 2-step performance. Specifically, we conduct an additional ablation study by comparing adaptive weighted loss (as used in MeanFlow), Cauchy loss, without adaptive weighting, and Cauchy loss with adaptive weighting. As shown in table below, we observe that adaptive weighted loss and Cauchy loss achieves similar performance, and adding adaptive weighting slightly improves the performance.
>
> \begin{array}{lcccc}
> \hline
> \text{Loss} & \text{1-step FID} & \text{1-step FDD}& \text{2-step FID} & \text{2-step FDD}\newline
> \hline
> \text{Lp Loss} & \text{19.6} & \text{533.4} & \text{17.5} & 468.2 \newline
> \text{Cauchy Loss w/o weightin} & \text{19.5} & \text{532.4} & \text{17.4} & \text{465.1} \newline
> \text{Cauchy Loss w/ weighting} & \text{19.3} & \text{531.6} & \text{17.3} & \text{461.1} \newline
> \hline
> \end{array}
>
> Also, we will add discussion and citation on [5] in our revised manuscript.
>
> (Continued)

---

> ### Author Response · Authors · 2025-11-21
>
> ---
> ### W3. Ablation on other Diffusion models.
>
> We believe that our approach not only applies to vanilla DiT architecture, but also applies to various diffusion transformer architectures. Note that the architectural improvement can be done in several ways, for instance, one can use improved architectures, e.g., using RMSNorm instead of LayerNorm, using SwiGLU FeedForward Network (SwiGLU FFN) instead of plain MLP for transformer block, where LightningDiT [6] have shown. In addition, one can leverage vision foundation model aligned VAE instead of REPA during flow model training, or use representation autoencoders (RAE) [7]. As we have shown that better flow models are more suitable in achieving better flow maps, those improvements would benefit in achieving better flow maps.
>
> Meanwhile, it is intriguing to explore whether our approach applies to Decoupled Diffusion Transformer (DDT) [2] architecture, as they explicitly decoupled the diffusion transformer similar to ours. As such, we conducted an ablation study on DDT architecture. To see how our approach applies to DDT models, we first trained a DDT-L/2 model (with RoPE, SwiGLU, and RMSNorm) on ImageNet 256 dataset for 400K steps, which achieves FID=7.89 (w/o CFG) and FID=1.75 (w/ CFG). Then we train both DDT-MF and DDT-DMF models initialized from pretrained DDT-L/2 for 100K steps with guidance applied. For implementation of DDT-DMF, the encoder blocks of DDT are conditioned with time $t$ and class embedding, and the decoder blocks are conditioned with time $r$ and the latents $z$ generated by encoder. For implementation of DDT-MF, we add $r$ embedding to both conditioning of encoder and decoder blocks. Interestingly, we find that DDT-DMF models are trained without instability, whereas the DDT-MF model suddenly collapses at the initial stage of training. Thus, we add QK normalization to the DDT-MF model to resolve the instability issue.
>
> The results are shown in Table below. We observe that the DMF model can be successfully applied to the DDT model, and it outperforms MF baseline on both 1-step and 2-step generation performance, which is consistent with the results for DiT architectures (Table 1).
> \begin{array}{lcccc}
> \hline
> \text{Model} & \text{1-step FID} & \text{1-step FDD}& \text{2-step FID} & \text{2-step FDD}\newline
> \hline
> \text{DDT-MF}  & \text{3.80} & \text{186.5} & \text{2.31} & \text{111.1} \newline
> \text{DDT-DMF} & \text{2.75} & \text{170.8} & \text{2.15} & \text{101.6} \newline
> \hline
> \end{array}
>
> We will add above experimental results and discussions in our revised manuscript.
>
> ---
> ### Q1. Representation quality.
>
> We find that representation quality has no big differences between MF and DMF models, as our fine-tuning objective does not incur any alignment during training. For instance, we measured the CKNNA between MF and DMF model hidden features using DINOv2-g following the REPA paper, yet there is no difference in its value considering various choices of $t,r$. In addition, we find that the CKNNA decreases when we conduct fine-tuning, e.g., CKNNA of DMF-XL/2+ at 8th layer is 0.48, while CKNNA of SiT-XL/2+REPA is 0.54. This is due to the fact that we did not use representation alignment loss during DMF fine-tuning.
>
> ---
> ### Q2. Using REPA during DMF fine-tuning.
> We find that using REPA did not show improvements when trained using model guidance (MG). On the other hand, we find that REPA improves the performance of DMF when compared to that which does not use MG. Specifically, we compare the DMF-XL/2+ and DMF-XL/2+REPA (which uses REPA during DMF fine-tuning) in the table below. Note that we trained each model for 200K iterations, and we use MG scale = 0.5 for DMF-XL/2+REPA, where MG scale=0.6 did show inferior results.
>
> \begin{array}{lccccc}
> \hline
> \text{Model} & \text{MG scale} & \text{1-step FID} & \text{1-step FDD}& \text{2-step FID} & \text{2-step FDD}\newline
> \hline
> \text{DMF-XL/2+} & \text{0.0} & \text{8.89} & \text{278.2} & \text{7.91} & \text{212.7} \newline
> \text{DMF-XL/2+REPA} & \text{0.0} &  \text{8.07} & \text{272.2} & \text{7.01} & \text{203.5} \newline
> \hline
> \text{DMF-XL/2+} & \text{0.6} & \text{2.25} & \text{122.3} & \text{1.69} & \text{74.6} \newline
> \text{DMF-XL/2+REPA} & \text{0.5} &  \text{2.49} & \text{151.2} & \text{1.73} & \text{91.3} \newline
> \hline
> \end{array}
>
> One can observe that while REPA enhances the performance when trained without MG, the gain diminishes if we apply MG during fine-tuning. We believe that choosing appropriate guidance during fine-tuning is more important than representation regularization when training high-performance flow maps, while representation regularization helps at the flow model pretraining stage.
>
> (Continued)

---

> > ### Author Response · Authors · 2025-11-21
> >
> > ---
> > ### Q3. Loss formulation.
> > First, we would like to clarify that our adaptive weighting function does not break the correctness of the FM objective. Specifically, if we do not use MF loss, the loss becomes  $\log (e^{\phi(t)} L_{\text{FM}}(\theta;t) + c) +\frac{\phi(t)}{2}$, which where the optimal solution also induces $v_\theta(x_t,t) = \mathbb{E}[v|x_t=x]$. While we have tried using Flow Matching loss as is (and use Cauchy loss for DMF loss), there were no big differences in terms of training stability or convergence, while we find the mismatch in loss scales lead to slightly inferior performance. As such, we use adaptive Cauchy loss for both FM loss and MF loss, which simply worked well in practice.
> >
> > ---
> > ### Q4. Timestep sampling.
> > For your information, we have tried such alternative timestep sampling, i.e., $p(t,r) = p(t)p(r|t)$, by sampling $t$ from Logit-Normal distribution, and sample $r$ from truncated logit-normal distribution, i.e., $r\sim LN(\mu, \sigma)$, but satisfying $r<t$. While we have tried various ablation on such a time proposal method,  we did not observe performance gain nor improvement in training stability. Since we found that using min-max strategy worked well in practice, we used it throughout the experiments.
> >
> > ---
> > ### Reference
> > [1] Chen, Junsong, Shuchen Xue, Yuyang Zhao, Jincheng Yu, Sayak Paul, Junyu Chen, Han Cai, Song Han, and Enze Xie. "Sana-sprint: One-step diffusion with continuous-time consistency distillation." arXiv preprint arXiv:2503.09641 (2025). \
> > [2] Dehghani, Mostafa, et al. "Scaling vision transformers to 22 billion parameters." International conference on machine learning. ICML, 2023. \
> > [3] Lu, Cheng, and Yang Song. "Simplifying, stabilizing and scaling continuous-time consistency models." arXiv preprint arXiv:2410.11081 (2024). \
> > [4] Wang, Shuai, Zhi Tian, Weilin Huang, and Limin Wang. "Ddt: Decoupled diffusion transformer." arXiv preprint arXiv:2504.05741 (2025). \
> > [5] Dao, Quan, Khanh Doan, Di Liu, Trung Le, and Dimitris Metaxas. "Improved training technique for latent consistency models." arXiv preprint arXiv:2502.01441 (2025). \
> > [6] Yao, Jingfeng, Bin Yang, and Xinggang Wang. "Reconstruction vs. generation: Taming optimization dilemma in latent diffusion models." CVPR, 2025. \
> > [7] Zheng, Boyang, et al. "Diffusion Transformers with Representation Autoencoders." arXiv preprint arXiv:2510.11690 (2025)

---

> > > ### Author Response · Authors · 2025-11-28
> > >
> > > Dear Reviewer 1Fiu,
> > >
> > > As the discussion period concludes in the coming week, we would like to offer a gentle reminder in case you have any remaining questions or concerns. We hope that our rebuttal has addressed your points, but please feel free to let us know if any clarification would be helpful.
> > >
> > > Thank you for your time and consideration.
> > >
> > > Sincerely,
> > > The Authors

---

### Official Review · Reviewer_9NpD · 2025-10-29

**Soundness:** 3
**Presentation:** 3
**Contribution:** 3
**Rating:** 6
**Confidence:** 3

**Summary:**

The paper proposes Decoupled MeanFlow (DFM), a method to accelerate diffusion or flow models by converting them into flow maps which can do large denoising jumps. DMF decouples the model's architecture into an encoder and a decoder, where only the encoder receives information about the start of the jump and only the decoder receives information about the end of the jump. DMF achieves the state-of-the-art performance on ImageNet, outperforming shortcut models, MeanFlow models and GANs.

**Strengths:**

- The proposed DMF model consistently outperforms the MeanFlow baseline across all evaluated datasets and variants. Notably, it achieves high-quality 1-step ImageNet generation which highlights its efficiency and strong generative capacity.
- The model can be trained from scratch, yet it also seamlessly integrates with existing pretrained models without requiring any architectural modifications while yielding improved results.
- The analysis of the encoder–decoder decomposition is interesting. It is particularly noteworthy that a pretrained model, when utilized under the proposed scheme and without any training, can outperform the original model in few-step generation regimes.

**Weaknesses:**

- While the separation of encoder and decoder components is appealing, it also seems natural to consider joint conditioning mechanisms that integrate information from both the current and target timesteps in some blocks, potentially via lightweight modifications such as joint AdaLN conditioning or LoRA adapters. Have the authors explored such hybrid alternatives?
- Given that MeanFlow already incorporates both timestep conditionings, one might expect the model to implicitly learn to attenuate or emphasize the relevant conditioning at different stages (e.g., down-weighting target timestep signals early and source signals later). Why does the explicit disentanglement in DMF lead to consistently better results? A more detailed discussion or theoretical analysis on this would strengthen the conceptual clarity of the paper.

**Questions:**

See weaknesses.

---

> ### Author Response · Authors · 2025-11-21
> **Response to Reviewer 9NpD**
>
> Dear reviewer 9NpD,
>
> We sincerely appreciate your valuable comments. We respond to your questions below.
>
> ---
> ### W1 & W2. Design choices.
>
>
> As you mentioned, the joint conditioning could also be possible, yet our simple design favors the advantages that 1) we do not need to modify the original flow model architecture, and 2) our approach provides better initialization as DMF is equivalent to flow model when $r=t$. On the other hand, adding timestep embedding MLP layers for $r$ (as MeanFlow baseline did), joint conditioning (e.g., concatenate $t$ and $r$ embeddings), and adapters (e.g., LoRA) inevitably modifies the original model. Note that from the definition of flow maps, letting $r=t$ becomes the original flow model, thus maintaining the model is helpful due to better initialization. Furthermore, from our experiments, the DMF shows performance gain compared to other all conditioning baseline (MeanFlow).
>
> In addition, regarding your suggestion, we provide additional experiments on such hybrid approaches. To see how addition of $r$ embeddings affect the few-step performance, we ablate onto following 4 design choices:
> - DMF (ours): Encode with $t$ + decode with $r$ without using additional $r$ embedder
> - MF-D: Encode with $t$ + Decode with $(t,r)$ using additional $r$ embedder
> - MF: Encode with $(t,r)$ + Decode with $(t,r)$ using additional $r$ embedder
> - MF-E: Encode with $(t,r)$ + Decode with $t$ using additional $r$ embedder
>
> Note that we use depth at 18 following Table 1, initialized from the SiT-L/2 pretrained for 80 epochs, and conduct flow map fine-tuning for 20 epochs without using guidance. The results are shown below:
>
> \begin{array}{lcccc}
> \hline
> \text{Method} & \text{1-step FID} & \text{1-step FDD}& \text{2-step FID} & \text{2-step FDD}\newline
> \hline
> \text{DMF}  & 19.3 & 531.6 & 17.3 & 461.1 \newline
> \text{MF-D} & 19.2 & 530.7 & 17.4 & 462.1 \newline
> \text{MF}   & 20.6 & 540.1 & 18.1 & 476.3 \newline
> \text{MF-E} & 20.8 & 548.7 & 17.7 & 465.3 \newline
> \hline
> \end{array}
>
>
> One can observe that using $r$ embedding to the encoder (e.g., MF and MF-E) achieves inferior performance than adding $r$ only to the decoder (e.g., DMF and MF-D) in both 1-step and 2-step generation. Furthermore, adding $r$ only to the decoder (MF-D) achieves comparable performance to DMF, which validates the importance of removing $r$ information at the encoder stage. We will append those ablation studies in our revised manuscript.
>
> Alternatively, as you mentioned, attenuating the impact of timestep conditionings (e.g., using routers to learn effective conditioning for each block) could be an interesting approach. It is still an open problem to analyze how each transformer block uses the timestep information during training, and we believe it is an interesting future work.

---

> > ### Author Response · Authors · 2025-11-28
> >
> > Dear Reviewer 9NpD,
> >
> > As the discussion period concludes in the coming week, we would like to offer a gentle reminder in case you have any remaining questions or concerns. We hope that our rebuttal has addressed your points, but please feel free to let us know if any clarification would be helpful.
> >
> > Thank you for your time and consideration.
> >
> > Sincerely,
> > The Authors

---

### Official Review · Reviewer_Rqpx · 2025-11-01

**Soundness:** 2
**Presentation:** 3
**Contribution:** 2
**Rating:** 4
**Confidence:** 5

**Summary:**

This paper mainly proposes an alternative time-conditioning design for MeanFlow models, named decoupled MeanFlow (DMF). Instead of modifying the architecture of a flow model to accommodate two timesteps,  DMF treats the DiT as an encoder-decoder architecture, and feed $t$ to the encoder and $r$ to the decoder, which leads to notable gains in generation quality. Combining this design with other technics such as flow matching warm-up, adaptive weighted Cauchy loss, and representation alignment, DMF achieves the state-of-the-art 1-step FID of 2.16 on ImageNet 256x256.

**Strengths:**

- The central contribution of this work is the simple time conditioning modification. This leads to notable performance gains as shown in the ablation studies in Table 1 and 2, validating the effectiveness of the decoupled design.
- Overall, a 1-step FID of 2.16 on ImageNet 256x256 is an impressive state-of-the-art result.

**Weaknesses:**

- This work is motivated by the encoder-decoder design. Yet the experiments does not provide direct evidence to prove that encoder-decoder is the key to high quality. For example, there are many other ways to condition the network without modifying its architecture, like interleaving $t$ and $r$ for the DiT blocks. Discussing alternative design choices in the ablation could strengthen the argument.
- The authors claim that an existing SiT can be converted into a flow map without finetuning, yet the results in Fig. 3 only show a very minor gap between DMF and SiT. Sometimes it's even worse than the baseline (Fig. 8 left). The slight improvement in general is unsurprising since the flow map tends to be closer to the average of start- and end-point velocity, thus it's reasonable that mixing start and end time in the model should be better than the plain SiT baseline.
- FM training with REPA is not really something new. It's not surprising that initializing from REPA models generally improves MF and DMF, which is not unique to DMF. Thus its relevance to the main contribution is weak.
- Writing issues:
  - Flow models trained on random noise-data pairs predicts the expectation of $\alpha_t^\prime x_0 + \sigma_t^\prime \epsilon$, i.e., $v(x, t) = \mathbb{E}[\alpha_t^\prime x_0 + \sigma_t^\prime \epsilon]$. The definition of $v(x, t)$ in L146 is not accurate.
  - Metric consistency: L422 says "DMF-XL/2 achieves 1-step FID=3.10", yet table 2 shows FID=2.83.

**Questions:**

In L208, it is stated that feeding the next timestep to both the encoder and decoder is redundant, which motivates DMF. Why would redundancy cause issues for the model? Neural networks are generally redundant by design, so I do not really understand the intuition of why feeding the timestep to selective blocks would be better than feeding it to all the blocks, apart from the concerns of architecture modifications.

---

> ### Author Response · Authors · 2025-11-21
> **Response to Reviewer Rqpx**
>
> Dear reviewer Rqpx,
>
> We sincerely appreciate your valuable comments. We address each of your questions and concerns one-by-one in what follows.
>
> ---
> ### W1. Ablation on the architectural design.
> We believe our experiments already have shown the evidence that the encoder–decoder design is the key mechanism behind DMF. First, note that converting a pretrained flow model into DMF without any fine-tuning (Figure 3a, 3b) achieves better performance. This observation indicates that flow models inherently learn to decouple into encoder-decoder schemes, and DMF explicitly aligns with this structure. Second, decoder-only fine-tuning (Figure 3c) further confirms this: the encoder can be frozen entirely, yet the decoder alone can learn accurate flow maps, which nearly matches the performance of full fine-tuning (8-step FID=1.68). This would not be possible if earlier blocks needed access to the next timestep, showing that the encoder’s role is to produce a stable $t$-conditioned representation and the decoder’s role is to transform it toward the next timestep $r$. Finally, the depth ablation (Table 1) demonstrates that performance depends strongly on where the decoder begins, again highlighting that the location of $r$-conditioning is crucial. Together, these results show that the encoder–decoder decoupling is not an arbitrary choice but the central factor that enables DMF to learn effective flow maps.
>
> Nonetheless, following your suggestions, we provide an additional experiment to examine how addition of $r$ conditioning affects the flow map learning performance. Specifically, we ablate onto following 4 design choices:
> - DMF (ours): Encode with $t$ + decode with $r$ without using additional $r$ embedder
> - MF-D: Encode with $t$ + Decode with $(t,r)$ using additional $r$ embedder
> - MF: Encode with $(t,r)$ + Decode with $(t,r)$ using additional $r$ embedder
> - MF-E: Encode with $(t,r)$ + Decode with $t$ using additional $r$ embedder
>
> Note that we use depth at 18 following Table 1, initialized from the SiT-L/2 pretrained for 80 epochs, and conduct flow map fine-tuning for 20 epochs without using guidance. The results are shown in the table below:
> \begin{array}{lcccc}
> \hline
> \text{Method} & \text{1-step FID} & \text{1-step FDD}& \text{2-step FID} & \text{2-step FDD}\newline
> \hline
> \text{DMF}  & 19.3 & 531.6 & 17.3 & 461.1 \newline
> \text{MF-D} & 19.2 & 530.7 & 17.4 & 462.1 \newline
> \text{MF}   & 20.6 & 540.1 & 18.1 & 476.3 \newline
> \text{MF-E} & 20.8 & 548.7 & 17.7 & 465.3 \newline
> \hline
> \end{array}
>
> One can observe that using $r$ embedding to the encoder (e.g., MF and MF-E) achieves inferior performance than adding $r$ only to the decoder (e.g., DMF and MF-D) in both 1-step and 2-step generation. Furthermore, adding $r$ only to the decoder (MF-D) achieves comparable performance to DMF, which validates the importance of removing $r$ information at the encoder stage. We will append those ablation studies in our revised manuscript.
>
> ---
> ### W2. Analyzing SiT vs DMF results in Figure 3.
> We would like to highlight that the results in Fig. 3 are not warranted. First, note that the flow models are only trained to predict the instant velocity, thus it is natural to expect that DMF performs worse than the original flow model. However, we show that if we choose appropriate layers to inject $r$ instead of $t$, there is a performance gain without changing the model weights. Thus, we believe this is surprising, and such observation motivates us to investigate such encoder-decoder design.
> Furthermore, we would like to clarify that the statement `a flow map tends to be close to the average of the start- and end-point velocity` refers to a theoretical characterization of the true probability-flow ODE, but not to the mixture of the model’s predictions at times $t$ and $r$. Specifically, it is true that the average velocity (i.e., flow map) can be approximated by the combination of the start and end point velocities as
> $$
> \frac{1}{r-t}\int_t^r v(x_\tau, \tau)d\tau \approx \frac{1}{2} (v(x_t,t) + v(x_r,r)).
> $$
> However, mixing the velocity prediction within the model layers, i.e., our encoder-decoder design, does not necessarily approximates such combination as
> $$
> \frac{1}{2} (g(f(x_t,t), t) + g(f(x_r,r), r)) \neq g(f(x_t,t), r),
> $$
> in general, where $f$ is an encoder (earlier blocks) and $g$ is a decoder (later blocks). Thus, our approach is not motivated by utilizing the characterization of probability flow ODE, but from the observation that the flow models have inherent structure that decouples the encoding and decoding scheme.
>
> (Continued)

---

> > ### Author Response · Authors · 2025-11-21
> >
> > ---
> > ### W3. Training with REPA.
> >
> > As for the role of REPA, our intention is not to claim REPA itself as a contribution, but rather to analyze how representation quality influences flow-map learning. As shown in Table 1, REPA improves both MF and DMF models, yet the improvement is larger for DMF. This indicates that DMF is able to leverage strong encoder representations more effectively than MF due to its design. Furthermore, our additional analysis in Appendix C shows that even the conversion itself (i.e., turning a pretrained flow model into a DMF model without fine-tuning) benefits meaningfully from REPA, which is consistent with our architectural intuition: since DMF relies heavily on the encoder that produces a representation $h_t$ before decoding toward timestep $r$, stronger representation learning directly translates into better flow-map quality. Thus, REPA is not introduced as a standalone novelty, but as evidence supporting the central claim of our work that representation quality is critical for flow maps, and DMF is particularly well-suited to exploit it.
> >
> > ---
> > ### W4. Writing issues
> > Thanks for the correction, we will update the draft to make it more precise in the descriptions of conditional flow matching objectives, and the typos regarding the metrics.
> >
> > ---
> > ### Q1. Justification on the design choices.
> > Our intention is not to argue that redundancy is inherently harmful, but to better leverage the structure of flow models to effectively learn flow maps from them. In particular, the location where the next timestep information is injected matters (which we have shown in W1). Early DiT blocks primarily encode $x_t$, thus having nothing to do with $r$, and later blocks determine how this representation should move toward $r$. Conditioning all blocks on both $t$ and $r$ entangles these roles and empirically hurts performance.
> > This is supported directly by our results: the MeanFlow baseline (conditioning every block on $(t,r)$) consistently underperforms DMF, and even converting a pretrained SiT into DMF without fine-tuning improves quality (Figure 3). This indicates that separating encoder(t) and decoder(r) aligns with the model’s inherent functional structure.
> > We will clarify this intuition in the final version.

---

> > > ### Author Response · Authors · 2025-11-28
> > >
> > > Dear Reviewer Rqpx,
> > >
> > > As the discussion period concludes in the coming week, we would like to offer a gentle reminder in case you have any remaining questions or concerns. We hope that our rebuttal has addressed your points, but please feel free to let us know if any clarification would be helpful.
> > >
> > > Thank you for your time and consideration.
> > >
> > > Sincerely,
> > > The Authors

---

### Meta-Review · Area_Chair_VkD4 · 2026-01-06

**Summary:**

The reviewers consistently acknowledge the strengths of this work, for example: (1) notable performance gains, (2) impressive state-of-the-art results, (3) conceptually elegant and practically impactful, and (4) comprehensive empirical validation. By carefully reviewing the authors' responses, I believe that the authors have addressed the major concerns of reviewers. Therefore, my recommendation for this submission is acceptance.

**Reviewer Concerns:**

The reviewers consistently acknowledge the strengths of this work, including (1) notable performance gains, (2) impressive state-of-the-art results, (3) conceptually elegant and practically impactful, and (4) comprehensive empirical validation. During the rebuttal, the authors have addressed the architectural concerns by providing sufficient ablation studies and analyses. The authors clarify that REPA is not introduced as a standalone novelty, but as evidence supporting the central claim, and demonstrate that turning a pretrained flow model into DMF without any fine-tuning already yields performance gains.

**Reviewer Scores:**

The initial reviewer scores are 6/6/4/6, with three reviews being positive. By carefully reviewing the reviewers’ comments and the authors’ responses, I believe that the reviewers would maintain an overall positive attitude toward this work after discussion.

---

### Decision · Program_Chairs · 2026-01-26

Accept (Poster)